# IgSF9b regulates anxiety behaviors through effects on centromedial amygdala inhibitory synapses

Olga Babaev[1,2], Hugo Cruces-Solis[1], Carolina Piletti Chatain[1,2], Matthieu Hammer[1], Sally Wenger[1], Heba Ali[1,2], Nikolaos Karalis[3], Livia de Hoz[4], Oliver M. Schlüter[5], Yuchio Yanagawa[6], Hannelore Ehrenreich[7], Holger Taschenberger[1], Nils Brose[1] & Dilja Krueger-Burg[1]

Abnormalities in synaptic inhibition play a critical role in psychiatric disorders, and accordingly, it is essential to understand the molecular mechanisms linking components of the inhibitory postsynapse to psychiatrically relevant neural circuits and behaviors. Here we study the role of IgSF9b, an adhesion protein that has been associated with affective disorders, in the amygdala anxiety circuitry. We show that deletion of IgSF9b normalizes anxiety-related behaviors and neural processing in mice lacking the synapse organizer Neuroligin-2 (Nlgn2), which was proposed to complex with IgSF9b. This normalization occurs through differential effects of Nlgn2 and IgSF9b at inhibitory synapses in the basal and centromedial amygdala (CeM), respectively. Moreover, deletion of IgSF9b in the CeM of adult Nlgn2 knockout mice has a prominent anxiolytic effect. Our data place IgSF9b as a key regulator of inhibition in the amygdala and indicate that IgSF9b-expressing synapses in the CeM may represent a target for anxiolytic therapies.

[1] Department of Molecular Neurobiology, Max Planck Institute of Experimental Medicine, 37075 Göttingen, Germany. [2] Göttingen Graduate School for Neurosciences, Biophysics, and Molecular Biosciences, 37077 Göttingen, Germany. [3] Friedrich Miescher Institute for Biomedical Research, 4058 Basel, Switzerland. [4] Department of Neurogenetics, Max Planck Institute of Experimental Medicine, 37075 Göttingen, Germany. [5] European Neuroscience Institute Göttingen, 37077 Göttingen, Germany. [6] Department of Genetic and Behavioral Neuroscience, Gunma University Graduate School of Medicine, Gunma, Maebashi 371-8511, Japan. [7] Clinical Neuroscience, Max Planck Institute of Experimental Medicine, 37075 Göttingen, Germany. These authors contributed equally: Olga Babaev and Hugo Cruces-Solis. Correspondence and requests for materials should be addressed to D.K.-B. (email: krueger@em.mpg.de)

Understanding the molecular basis of psychiatric disorders is one of the foremost medical challenges of our times, and substantial interest has arisen in the role of synaptic dysfunction in psychiatric pathophysiology[1,2]. The vast majority of corresponding studies have focused on glutamatergic, excitatory synapses, but it is becoming increasingly clear that an equally important contribution stems from abnormalities in inhibitory synaptic transmission[3–8]. This is particularly true for anxiety disorders, which have been linked to decreased inhibition in multiple brain regions[9–11]. To date however, astonishingly little is known about the molecular mechanisms by which abnormalities at inhibitory synapses contribute to the pathophysiology of anxiety disorders.

A primary reason for this dearth of information is that the molecular composition of inhibitory postsynapses is only just being uncovered. In the past 5 years, the list of known inhibitory synapse organizers has expanded dramatically due to major technological advances[12]. A key question now is whether these molecules function uniformly across inhibitory synapses, or whether they display the same heterogeneity of function that is observed at the cellular level in the inhibitory network[3,13]. Understanding this organization will not only be essential for identifying disease mechanisms in pathological anxiety and other disorders, but may also provide a unique opportunity for uncovering selective targets for circuit-specific therapeutic interventions.

A particularly interesting candidate molecule is IgSF9b, a member of the immunoglobulin superfamily of cell adhesion proteins that was recently shown to localize to inhibitory synapses in dissociated neuron cultures[14–16]. Variants in IgSF9b are associated with major depression and the negative symptoms of schizophrenia[17–19], indicating that IgSF9b may regulate affect in human patients. IgSF9b was recently reported to act in a complex with Neuroligin-2 (Nlgn2)[14], a member of the Neuroligin family of adhesion proteins specifically found at inhibitory synapses[12,20–22]. Loss-of-function mutations in Nlgn2 have been identified in patients with schizophrenia or pathological anxiety and autism[23,24], and in mice, deletion of Nlgn2 robustly enhances anxiety-related behaviors[8,25]. Together, these findings raise the intriguing possibility that IgSF9b, like Nlgn2, may contribute to psychiatric pathophysiology by regulating synaptic inhibition in behaviorally relevant circuitry. To date, however, the molecular, cellular and circuitry functions of IgSF9b and its potential interactions with Nlgn2 in vivo remain completely unexplored.

In the present study, we investigated the role of synaptic inhibition in anxiety by addressing two key questions: First, does IgSF9b regulate anxiety-related behavior and neural processing in the anxiety circuitry of WT and/or pathologically anxious Nlgn2 KO mice? Second, do IgSF9b and/or Nlgn2 act at specific inhibitory synapses within the anxiety circuitry and thus offer synapse-specific targets for interventions? To this end, we investigated the consequences of IgSF9b deletion and IgSF9b x Nlgn2 double deletion on anxiety-related behavior and circuits, as well as on inhibitory synapses in the amygdala, a brain region that plays a central role in the processing of anxiety information[9,10,26,27]. We show that IgSF9b regulates anxiety-like behaviors in Nlgn2 KO mice and that IgSF9b and Nlgn2 exert differential effects on distinct components of the amygdala inhibitory circuitry. Our data provide the first characterization of IgSF9b function in vivo and uncover a prominent anxiolytic consequence of IgSF9b deletion in Nlgn2 knockout (KO) mice, implicating IgSF9b-expressing neurons in the centromedial (CeM) amygdala as potential targets for anxiolytic therapies.

## Results

**Deletion of IgSF9b normalizes anxiety in Nlgn2 KO mice.** To address the role of IgSF9b in anxiety processing, we first tested whether IgSF9b deletion would alter anxiety-related behaviors and/or modulate the anxiety phenotype observed in Nlgn2 KO mice. To this end, we assessed the behavior of male adult WT, Nlgn2 single KO, IgSF9b single KO, and Nlgn2 x IgSF9b double KO mice in an open field (OF) apparatus (Fig. 1a–c). As expected[8,25], Nlgn2 KO mice spent significantly less time, traveled less distance, and made fewer entries into the anxiogenic center of the OF (Fig. 1d–g, red bars; ANOVA comparisons listed in Table 1; error bars represent SEM), indicative of enhanced anxiety-related behavior. Based on previous findings indicating that IgSF9b and Nlgn2 are components of the same molecular complex at inhibitory synapses[14], we expected that deletion of IgSF9b may increase anxiety and/or exacerbate the phenotype of Nlgn2 KO mice. To our surprise, however, IgSF9b KO mice demonstrated a trend towards decreased anxiety (Fig. 1e, f, Supplementary Fig. 1d, e, blue bars). Most strikingly, the prominent anxiety phenotype of the single Nlgn2 KO mice was completely abolished in the double KO mice, with all measures unchanged from WT levels (Fig. 1d–g, Supplementary Fig. 1a–h, purple bars). An identical pattern of behavior was observed in female mice (Supplementary Fig. 1i–m). Moreover, Nlgn2 KO mice spent less time, traveled less distance and made fewer entries into the anxiogenic open arms of the elevated plus maze (EPM) (Fig. 1h–m, red bars), while IgSF9b deletion abolished the anxiety phenotype of Nlgn2 KO mice (Fig. 1h–m, purple bars).

Interestingly, IgSF9b KO mice showed a significant hyperactivity in both the OF and the EPM (Fig. 1g, m, blue bar) that was not a result of a basic locomotor phenotype (Supplementary Fig. 2a–i). This increased exploratory activity may be an antagonistic behavior to freezing typically demonstrated by anxious mice in novel environment[28], and indeed "anxious" Nlgn2 KO mice demonstrate reduction in locomotor activity specifically under anxiogenic conditions (Fig. 1g, red bar; see also Babaev et al., 2016[8]). Taken together, these findings indicate a robust anxiolytic effect of IgSF9b deletion in Nlgn2 KO mice.

**Regulation of the anxiety circuitry by IgSF9b and Nlgn2.** To understand the mechanism underlying this behavioral normalization, we sought to identify the relevant neural circuits using immunostaining for the immediate early gene cFos as a marker for neuronal activity[29]. We focused on the amygdala due to its well-established central role in the processing of anxiety information[9,26,27] (Fig. 2a). Consistent with our previous report[8], neurons in the basal (BA) but not the LA were overactivated in Nlgn2 KO mice following exposure to the OF (Fig. 2a, left panels, red bars). However, this overactivation was not reversed in double KO mice (Fig. 2a, lower left panel, purple bar), and a trend towards an increase in activation in the IgSF9b KO was observed (Fig. 2a, lower left panel, blue bar). The observed increase in cFos was induced by exposure to the OF, as BA cFos in mice taken directly from their home cages did not differ among genotypes (Fig. 2b). Co-immunostaining with the interneuron markers parvalbumin (PV) and somatostatin (SOM) showed that the increase in double KO mice may be partially due to an increase in the number of activated PV-positive interneurons in these mice (Supplementary Fig. 3a–h).

BA neurons project to the centrolateral (CeL) and the CeM nucleus (Fig. 2a). Given that the CeM represents the primary output nucleus of the amygdala, which projects directly to downstream targets to mediate anxiety-related behaviors[9,27], we tested whether Nlgn2 and IgSF9b deletion may have opposite effects on CeM activation that underlie their opposing anxiety-

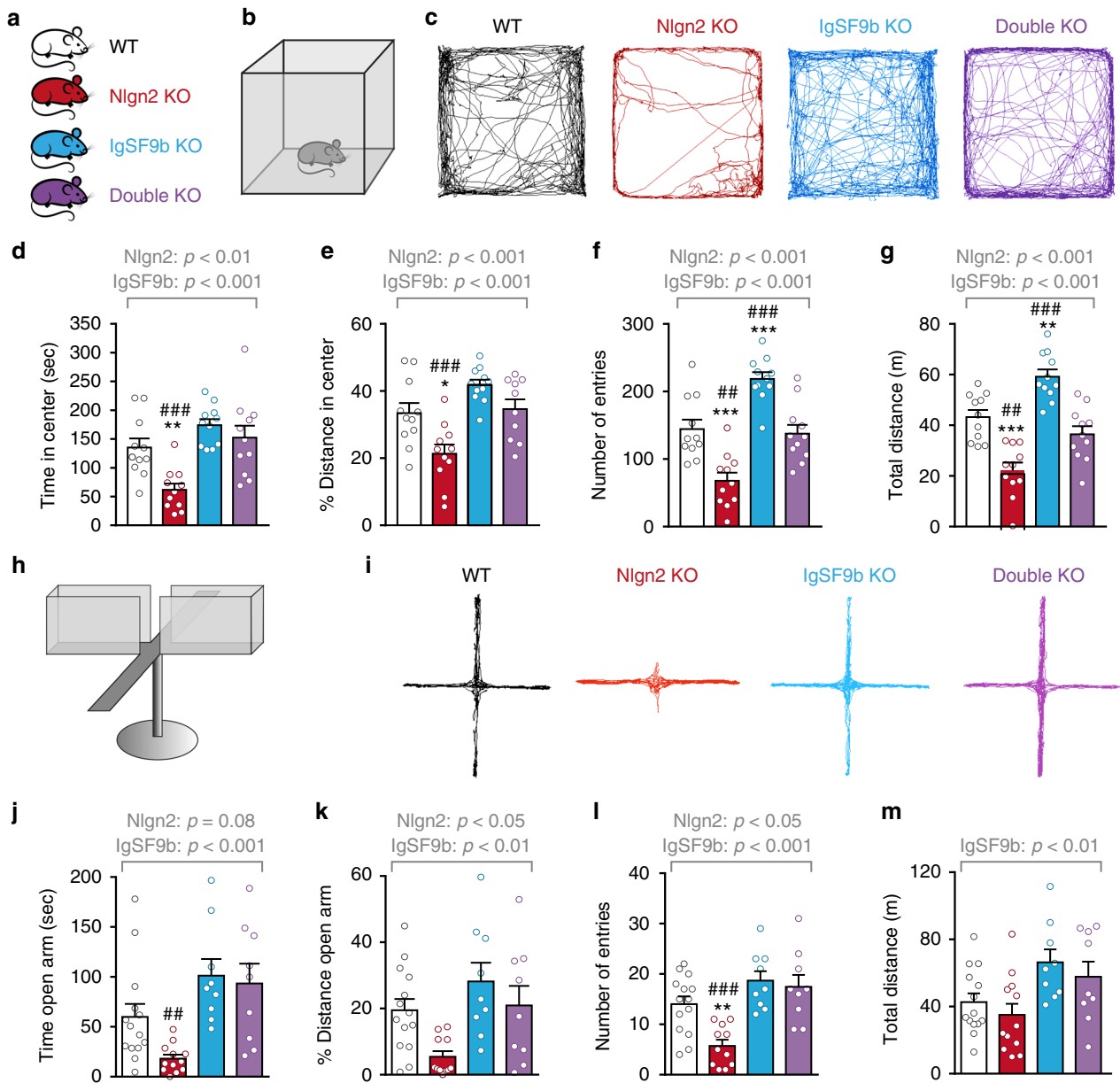

**Fig. 1** Deletion of IgSF9b normalizes anxiety-related behavior in Nlgn2 KO mice. **a**, **b** WT, Nlgn2 KO, IgSF9b KO, and double KO mice were assessed for anxiety-related behavior in an OF test. **c** Representative tracks of OF exploration. **d** Time spent in the anxiogenic center of the OF. **e** Distance traveled in the center of the OF, expressed as percentage of total distance traveled. **f** Number of entries into the center of the OF. **g** Total distance traveled in the OF. **h** Mice were assessed for anxiety-related behavior in EPM. **i** Representative tracks of EPM exploration. **j** Time spent in the anxiogenic open arms of the EPM. **k** Distance traveled in the open arms of the EPM, expressed as percentage of total distance traveled. **l** Number of entries into the open arms of the EPM. **m** Total distance traveled in the EPM. Statistically significant ANOVA comparisons are marked in gray at the top of the panels and are listed in Table 1. For all other ANOVA comparisons, $F < 1$. Post-hoc analysis: $*p < 0.05$ relative to WT, $**p < 0.01$ relative to WT, $***p < 0.001$ relative to WT, $\#p < 0.05$ relative to double KO, $\#\#p < 0.01$ relative to double KO, $\#\#\#p < 0.001$ relative to double KO. Error bars represent SEM, $n = 10$–12 mice per genotype for OF, $n = 9$–14 mice per genotype for EPM

related consequences. Analysis of CeM activation during OF exposure revealed an overactivation of the CeM in Nlgn2 KO mice (Fig. 2a, lower right panel, red bar), with no effect in IgSF9b KO mice (Fig. 2a, lower right panel, blue bar). Strikingly, the overactivation of the CeM was completely normalized in double KO mice (Fig. 2a, lower right panel, purple bar), which is highly consistent with their WT-like anxiety-related behavior. Similar analysis of the CeL revealed no effect in either Nlgn2 KO or IgSF9b KO mice (Fig. 2a, upper right panel).

To confirm that suppression of BA → CeM projection neurons is indeed the mechanism behind suppression of anxiety-related

activity in the CeM, we stereotaxically injected a fluorescent Retrobead retrograde tracer into the CeM to label neuronal populations that project directly to the CeM (Fig. 2c and Supplementary Fig. 3i-m). We measured activation of Retrobead-positive neurons in the BA under anxiogenic conditions using cFos immunohistochemistry (Fig. 2d). Importantly, the stereotaxic surgery did not alter anxiety in any of the KO mice (Supplementary Fig. 3l–m). Analysis of Retrobead- and cFos-double positive neurons revealed that BA → CeM direct projections were strongly overactivated in Nlgn2 KO mice (Fig. 2d, red bar), and surprisingly, overactivation persisted in double KO mice

**Table 1 Two-way ANOVA comparisons**

| Figure | Main effect Nlgn2 KO $F$-value | Main effect Nlgn2 KO $p$-value | Main effect IgSF9b KO $F$-value | Main effect IgSF9b KO $p$-value | Nlgn2 KO x IgSF9b KO interaction $F$-value | Nlgn2 KO x IgSF9b KO interaction $p$-value |
|---|---|---|---|---|---|---|
| 1d | $F_{1,40} = 10.02$ | 0.003 | $F_{1,40} = 18.58$ | <0.001 | $F_{1,40} = 2.98$ | 0.092 |
| 1e | $F_{1,39} = 12.00$ | 0.001 | $F_{1,39} = 17.60$ | <0.001 | $F_{1,39} < 1$ | n.s. |
| 1f | $F_{1,40} = 40.80$ | <0.001 | $F_{1,40} = 34.12$ | <0.001 | $F_{1,40} < 1$ | n.s. |
| 1g | $F_{1,40} = 53.93$ | <0.001 | $F_{1,40} = 25.73$ | <0.001 | $F_{1,40} < 1$ | n.s. |
| 1j | $F_{1,40} = 3.22$ | 0.080 | $F_{1,40} = 17.77$ | <0.001 | $F_{1,40} = 1.51$ | 0.227 |
| 1k | $F_{1,39} = 6.32$ | 0.016 | $F_{1,39} = 8.14$ | 0.007 | $F_{1,39} < 1$ | n.s. |
| 1l | $F_{1,39} = 7.29$ | 0.010 | $F_{1,39} = 21.70$ | <0.001 | $F_{1,39} = 4.02$ | 0.052 |
| 1m | $F_{1,40} = 1.33$ | 0.255 | $F_{1,40} = 11.04$ | 0.002 | $F_{1,40} < 1$ | n.s. |
| 2a (BA) | $F_{1,28} = 7.58$ | 0.010 | $F_{1,28} = 4.58$ | 0.041 | $F_{1,28} < 1$ | n.s. |
| 2a (CeM) | $F_{1,19} = 7.69$ | 0.012 | $F_{1,19} = 3.33$ | 0.084 | $F_{1,19} = 1.35$ | 0.259 |
| 2d | $F_{1,25} = 9.68$ | 0.005 | $F_{1,25} = 2.10$ | 0.160 | $F_{1,25} = 1.10$ | 0.305 |
| 2e | $F_{1,21} = 2.19$ | 0.154 | $F_{1,21} < 1$ | n.s. | $F_{1,21} = 1.04$ | 0.319 |
| 3e | $F_{1,17} < 1$ | n.s. | $F_{1,17} = 4.90$ | <0.05 | $F_{1,17} = 4.29$ | 0.054 |
| 3l | $F_{1,277} = 18.73$ | <0.001 | $F_{1,277} = 33.75$ | <0.001 | $F_{1,277} < 1$ | n.s. |
| 4f | $F_{1,23} < 1$ | n.s. | $F_{1,23} = 6.92^a$ | $0.015^a$ | $F_{1,23} = 5.37^a$ | $0.030^a$ |
| 4g | $F_{1,24} < 1$ | n.s. | $F_{1,24} = 6.20^a$ | $0.020^a$ | $F_{1,24} = 5.64^a$ | $0.026^a$ |
| 5d | $F_{1,48} = 1.83$ | 0.182 | $F_{1,48} = 1.22$ | 0.274 | $F_{1,48} < 1$ | n.s. |
| 5e | $F_{1,48} = 3.69$ | 0.061 | $F_{1,48} < 1$ | n.s. | $F_{1,48} = 4.67$ | 0.036 |
| 5h | $F_{1,72} = 1.27$ | 0.263 | $F_{1,72} = 6.25$ | 0.015 | $F_{1,72} = 1.33$ | 0.253 |
| 5j | $F_{1,71} = 6.83$ | 0.011 | $F_{1,71} < 1$ | n.s. | $F_{1,71} < 1$ | n.s. |
| 6c | $F_{1,28} < 1$ | n.s. | $F_{1,28} = 5.07$ | 0.032 | $F_{1,28} < 1$ | n.s. |
| 6d | $F_{1,20} < 1$ | n.s. | $F_{1,20} = 1.35$ | 0.259 | $F_{1,20} < 1$ | n.s. |
| 6f | $F_{1,26} < 1$ | n.s. | $F_{1,26} = 1.12$ | 0.300 | $F_{1,26} = 2.56$ | 0.122 |
| 6g | $F_{1,21} = 7.73$ | 0.011 | $F_{1,21} = 6.38$ | 0.02 | $F_{1,21} < 1$ | n.s. |
| 6i | $F_{1,75} = 14.61$ | 0.0003 | $F_{1,75} = 6.60^a$ | $0.012^a$ | $F_{1,75} = 3.43^a$ | $0.068^a$ |
| 6j | $F_{1,75} < 1$ | n.s. | $F_{1,75} = 11.41^a$ | $0.001^a$ | $F_{1,75} < 1^a$ | n.s. |
| 7c | $F_{1,60} = 11.26$ | 0.001 | $F_{1,60} < 1$ | n.s. | $F_{1,60} < 1$ | n.s. |
| 7e | $F_{1,60} = 14.09$ | 0.0004 | $F_{1,60} < 1$ | n.s. | $F_{1,60} = 2.54$ | 0.116 |
| 7h | $F_{1,24} = 7.19$ | 0.013 | $F_{1,24} < 1$ | n.s. | $F_{1,24} = 1.23$ | 0.278 |
| 7j | $F_{1,21} = 13.25$ | 0.002 | $F_{1,21} < 1$ | n.s. | $F_{1,21} < 1$ | n.s. |

n.s., not significant
$^a$Refers to IgSF9b shRNA rather than IgSF9b KO

(Fig. 2d, purple bar). In contrast, activation of inhibitory CeL → CeM projections did not differ among any of the genotypes (Fig. 2e). Together, these data indicate that Nlgn2 and IgSF9b deletion affect distinct targets within the amygdala: While Nlgn2 regulates anxiety-induced activation of projection neurons in BA, IgSF9b may normalize the CeM anxiogenic output by local mechanism within the CeM.

**IgSF9b and Nlgn2 bidirectionally regulate CeM activity**. To confirm that changes in the neural activity of the CeM accompany anxiety-related behavior, and to investigate whether IgSF9b and/or Nlgn2 alter specific neural substrates in the CeM in vivo, we recorded local field potential (LFP) oscillations in the CeM in freely moving mice during exploration of the OF (Fig. 3a, b and Supplementary Fig. 4a-b). Spectral analysis of CeM LFPs revealed that exposure to the OF increased oscillatory activity in all genotypes compared to home cage CeM activity, particularly in the theta (4–12 Hz) and beta (18–30 Hz) ranges (Supplementary Fig. 4b-c). The increase in the beta frequency band was most prominent in Nlgn2 KO mice, indicating that Nlgn2 deletion modifies anxiety-related CeM activity in the beta frequency range. To further explore whether this increase in beta power was modulated by anxiogenic conditions, we compared CeM activity during exploration (defined as speed > 1 cm/s) of the relative safety of the periphery with the potentially anxiogenic center of the OF (Fig. 3c, d). Beta power increased in Nlgn2 KO mice during exploration of the OF center, and this increase was

completely abolished in double KO mice (Fig.3e, Supplementary Fig. 4d). Furthermore, we found a significant correlation between the magnitude in beta power and the distance from the center of the OF specifically in Nlgn2 KO mice (Fig. 3f–i). The increase in beta power was not induced by changes in locomotion, since beta activity was not modulated by speed in any genotype (Supplementary Fig. 4e). These results indicate that deletion of Nlgn2 increases beta oscillatory activity in the CeM under anxiogenic conditions, while deletion of IgSF9b normalizes anxiety-related beta activity in double KO mice (Fig. 3d, e), a mechanism that may underlie the normalization of anxiety-like behavior.

To further characterize the increase in beta oscillatory activity in Nlgn2 KO mice and the normalization in double KO mice, we analyzed the temporal dynamics in beta power during transitions from the periphery to the center of the OF. In agreement with their robust avoidance of the center of the OF, the number of transitions of Nlgn2 KO mice was too low to perform a powerful statistical analysis (4.4 ± 2.6 transitions). However, we noticed that Nlgn2 KO mice displayed frequent stretch-attend postures (SAPs), a risk-assessment behavior that reflects an internal conflict between anxiety and the exploratory drive[30,31] (Fig. 3j). Nlgn2 KO mice showed a significant increase in the magnitude of beta power during SAPs compared to WT mice, indicative of a correlation between beta oscillations and anxiogenic conditions (Fig. 3k–l, red bar). Strikingly, IgSF9b showed a significant reduction in beta power, while double KO mice showed similar levels as WT mice (Fig. 3l, purple bar). Together, these data highlight that (1) CeM neural activity is modulated by the

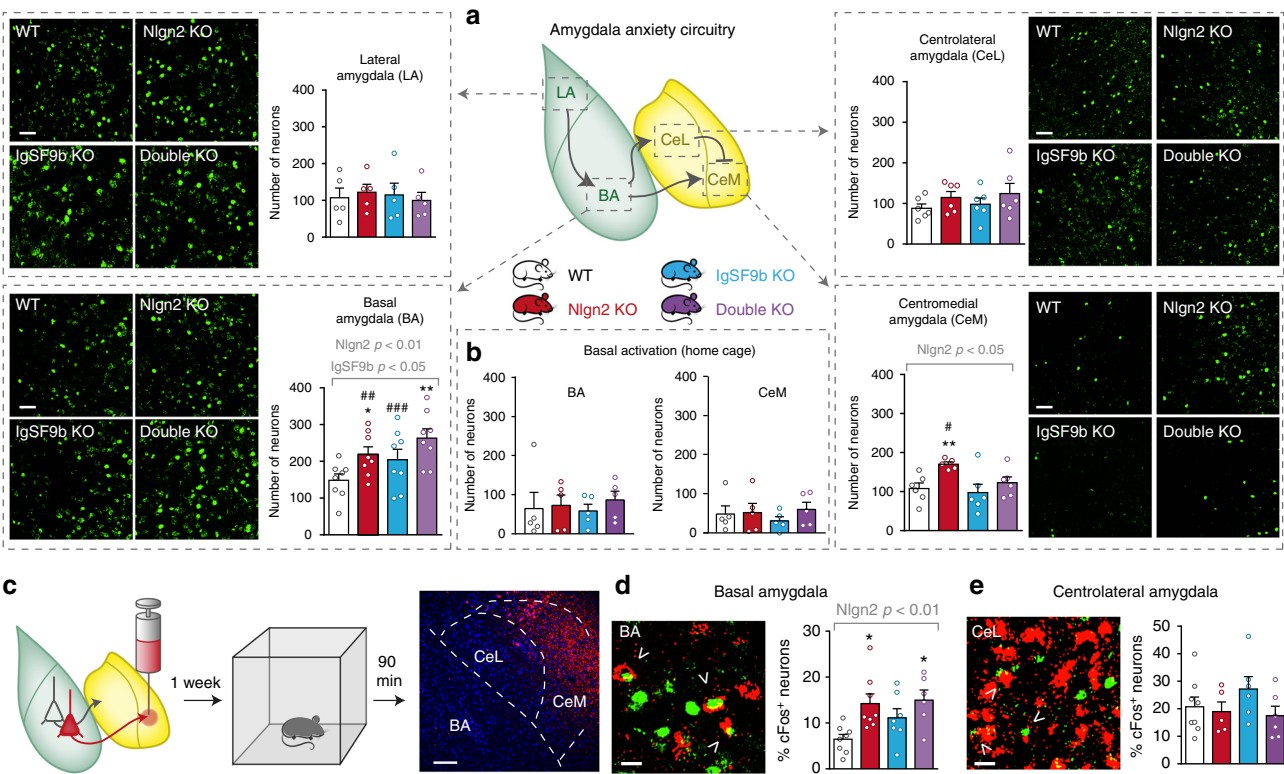

**Fig. 2** Deletion of IgSF9b normalizes neuronal activation in the CeM of Nlgn2 KO mice. **a** cFos immunolabeling in the LA, BA, CeL, and CeM in all four genotypes. Representative images and quantification of cFos-positive cells in the LA (upper left panel), the BA (lower left panel), the CeL (upper right panel), and the CeM (lower right panel). Scale bar, 50 μm. **b** Number of cFos-positive cells in the BA and CeM in WT, Nlgn2 KO, IgSF9b KO, and double KO mice under home cage conditions. **c** Experimental design of retrograde labeling. Mice were injected with Retrobeads into the CeM, and 1 week later they were exposed to the OF for 10 min, perfused 90 min later and analyzed for double labeling of cFos and Retrobeads in the BA and CeL. Scale bar, 100 μm. **d** Representative image and quantification of cFos- and Retrobead-double positive cells in BA in all four genotypes. Scale bar, 20 μm. **e** Representative image and quantification of cFos- and Retrobead-double positive cells in the CeL in all four genotypes. Scale bar, 20 μm. Statistically significant ANOVA comparisons are marked in gray at the top of the panels and are listed in Table 1. For all other ANOVA comparisons, $F < 1$. Post-hoc analysis: *$p < 0.05$ relative to WT, **$p < 0.01$ relative to WT, #$p < 0.05$ relative to double KO, ##$p < 0.01$ relative to double KO, ###$p < 0.001$ relative to double KO. Error bars represent SEM, $n = 5$–8 mice per genotype. WT, white bars; Nlgn2 KO, red bars; IgSF9b KO, blue bars; double KO, purple bars

valence of the environment ("safe" vs "anxiogenic"), (2) beta oscillatory activity in the CeM represents a novel neural signature of pathological anxiety induced by deletion of Nlgn2, and (3) Nlgn2 and IgSF9b bidirectionally modulate anxiety-related neural activity in the CeM particularly during risk-assessment behavior.

**IgSF9b knockdown in CeM normalizes anxiety in Nlgn2 KO mice**. To further confirm that IgSF9b acts in the CeM to modulate anxiety, and to investigate whether targeting IgSF9b-containing synapses in the adult amygdala may recapitulate the anxiolytic effects in the constitutive global KO, we locally reduced IgSF9b levels by adeno-associated virus (AAV)-mediated expression of IgSF9b shRNA (Fig. 4a–b). AAV particles encoding IgSF9b shRNA or control shRNA (a mutant construct that lacks knockdown activity, as reported previously[14]) were injected into the CeM of 8–12-week-old WT and Nlgn2 KO mice using stereotaxic surgery (Fig. 4c–d and Supplementary Fig. 5a), generating four experimental groups (Fig. 4e).

One day before and 6 weeks after surgery, anxiety-related behavior was assessed in an OF (Fig. 4c). IgSF9b shRNA had no significant effect on either the time or the distance traveled in the center of the OF in WT mice (Fig. 4f–g, white vs. blue and white shaded bars). In contrast, Nlgn2 KO mice injected with IgSF9b shRNA showed a pronounced reduction of anxiety-related behaviors compared to Nlgn2 KO mice injected with control

shRNA, as evidenced by a significant increase in both the time and the distance traveled in the center (Fig. 4f–g, red vs. red and blue shaded bars, and Fig. 4h, representative traces). Interestingly, anxiety levels appeared to be exacerbated in Nlgn2 KO but not in WT mice over the observed 6-week time period, even in mice that had not undergone surgery (Supplementary Fig. 5b-d), and this effect was completely reversed by local reduction of IgSF9b (Fig. 4f–h, Supplementary Fig. 5e-f). Given that IgSF9b is highly expressed in the CeM (Supplementary Fig. 6a-d), these data confirm that IgSF9b modulates anxiety-related behaviors through a CeM-specific mechanism, and indicates that targeting inhibitory transmission in the CeM can ameliorate anxiety-related behaviors.

**IgSF9b deletion alters CeM inhibitory synapse function**. To elucidate the synaptic mechanisms that may be responsible for the anxiolytic effect of IgSF9b deletion in the CeM, we recorded miniature inhibitory postsynaptic currents (mIPSCs) from acute brain slices obtained from adult mice (Fig. 5). To assist in the identification of the relevant brain structures and to distinguish between excitatory and inhibitory neurons, we used mice of all four genotypes that additionally expressed the fluorescent marker Venus under a vesicular inhibitory amino acid transporter (VIAAT) promoter[32]. Given that CeM neurons consisted primarily of inhibitory (Venus-positive) neurons, we restricted our analysis to these neurons.

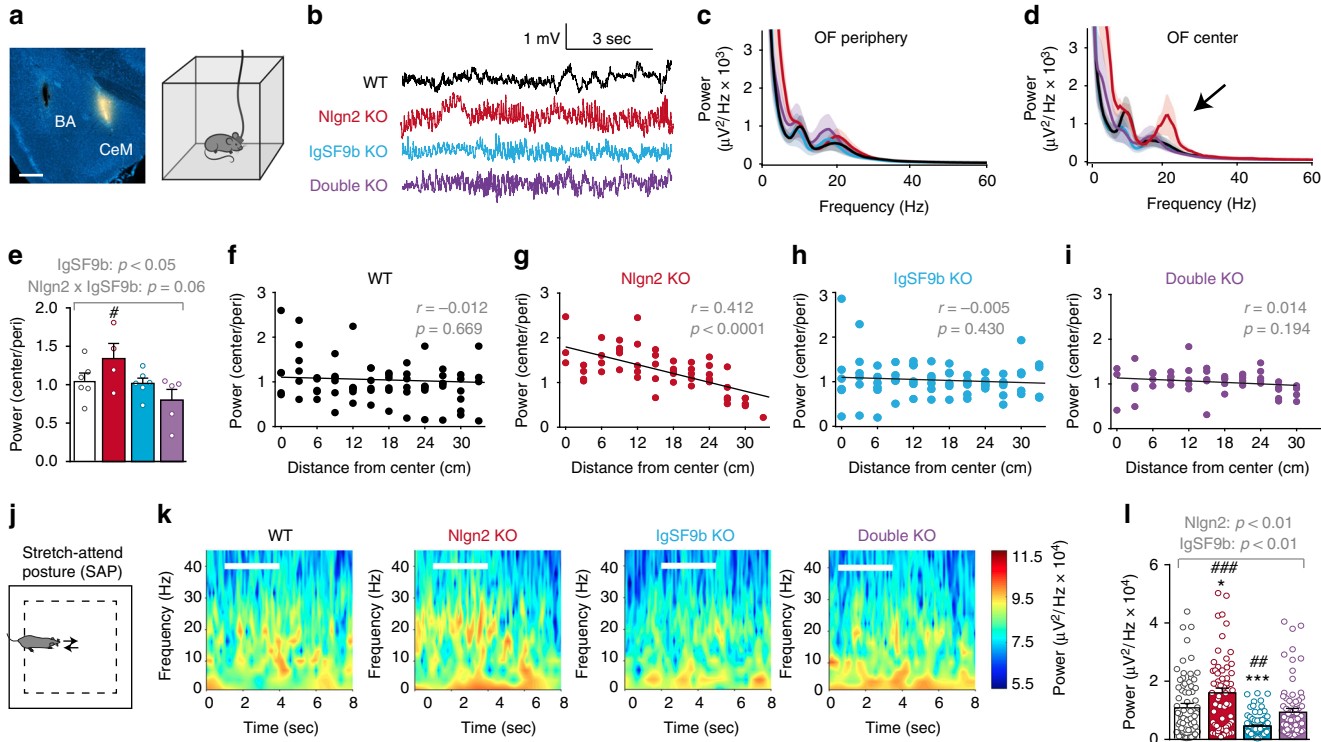

**Fig. 3** Neuronal activity in the beta frequency range is normalized in the CeM of double KO mice. **a** Representative image of the location of electrode in the CeM and experimental design. Scale bar, 500 μm. After electrode implantation mice were exposed to the OF for 15 min. **b** Representative traces of CeM LFPs from all four genotypes during SAP. **c**, **d** LFP power spectrum for all four genotypes during movement in the periphery (**c**) and center (**d**) of the OF. **e** Average power increase in the center relative to the periphery of the OF for the beta frequency range (18–30 Hz). **f–i** Increase in normalized beta power as a function of distance from the center in WT (**f**), Nlgn2 KO (**g**), IgSF9b KO (**h**), and double KO (**i**) mice. Statistical analysis of correlations is indicated in the figure. **j** Schematic of the stretch-attend posture (SAP) scored during the OF test. **k** Representative wavelet transforms of CeM LFP of the four genotypes during exploration of the OF. White horizontal lines indicate the duration of a representative SAP. **l** Average beta power during SAP. Statistically significant ANOVA comparisons are marked in gray at the top of the panels and are listed in Table 1. For all other ANOVA comparisons, $F < 1$. Post-hoc analysis: $*p < 0.05$ relative to WT, $***p < 0.001$ relative to WT, $\#p < 0.05$ relative to double KO, $\#\#p < 0.01$ relative to double KO, $\#\#\#p < 0.001$ relative to double KO. SAPs and mice: WT $n = 77/ 6$ mice; Nlgn2 KO $n = 62/ 5$ mice; IgSF9b KO $n = 84/ 6$ mice; double KO $n = 71/ 5$ mice. Error bars represent SEM. WT, white bars; Nlgn2 KO, red bars; IgSF9b KO, blue bars; double KO, purple bars

Neurons in the CeM showed typical firing patterns as previously reported (Fig. 5a, b)[33], and all parameters that reflect membrane excitability were similar between groups (Fig. 5c–e and Table 2). Surprisingly, however, IgSF9b deletion significantly increased mean mIPSC frequency while leaving mean mIPSC amplitudes unaffected (Fig. 5f–k, blue bars). This increase was due to a subset of IgSF9b KO neurons with substantially larger mean mIPSC frequencies, as revealed by a significant shift in the distribution of mean mIPSC frequencies among the neurons tested (Fig. 5i, blue vs. black line). Double KO mice showed a similar albeit less pronounced phenotype, with a trend towards an increase in mean mIPSC frequency (Fig. 5h, purple bar) and a significant shift in the distribution of mean mIPSC frequencies in the neuronal population (Fig. 5i, purple vs. black line). Consistent with our previous findings[8], Nlgn2 deletion did not affect mean mIPSC frequency, and only modestly reduced mean mIPSC amplitude (Fig. 5h–k, red bars). No effects of any genotype were observed on the frequency and amplitude of spontaneous postsynaptic excitatory currents (sEPSCs) in the CeM (Fig. 5l–q), although, interestingly, comparison of the probability distribution of mean frequencies among all cells to the corresponding WT distribution (Fig. 5o) revealed a small shift towards lower mean frequencies in double KO. Rise time and decay time of mIPSCs and sEPSCs were not significantly altered (Table 2). Together, our observations indicate that deletion of IgSF9b specifically alters the function of inhibitory, but not excitatory, synapses. These findings indicate

that IgSF9b deletion suppresses neuronal overactivation in the CeM of double KO mice, and hence may reduce their anxiety-related behavior, through increased inhibitory input onto CeM projection neurons.

**IgSF9b deletion increases VIAAT puncta in the CeM.** We next investigated the molecular mechanisms by which IgSF9b deletion increases synaptic inhibition in the CeM. Immunohistochemistry for IgSF9b revealed punctate staining in the BA and CeM of adult WT mice, which was absent in IgSF9b KO mice (Supplementary Fig. 6a–d). Co-staining of IgSF9b with Nlgn2 revealed that in the CeM, but not in the BA, these proteins are localized in adjacent overlapping domains (Supplementary Fig. 6e–f), consistent with previous findings in hippocampal cultures[14]. These data indicate that IgSF9b and Nlgn2 likely function in separate subdomains of the inhibitory synapse as previously proposed[14].

To assess the molecular basis of the increase in inhibitory transmission in IgSF9b KO and double KO mice, we performed immunohistochemistry for several markers of inhibitory synapses, including VIAAT, which labels inhibitory presynaptic terminals; gephyrin, the core scaffolding protein at inhibitory postsynapses; S-SCAM, a scaffolding protein that is only known synaptic interaction partner of IgSF9b; and the GABA$_A$R subunits γ2 and α1 (Fig. 6a–g, Supplementary Fig. 6g)[12,14]. Strikingly, IgSF9b KO and double KO mice showed a significant increase in the number of VIAAT puncta (Fig. 6c, blue and purple bars,

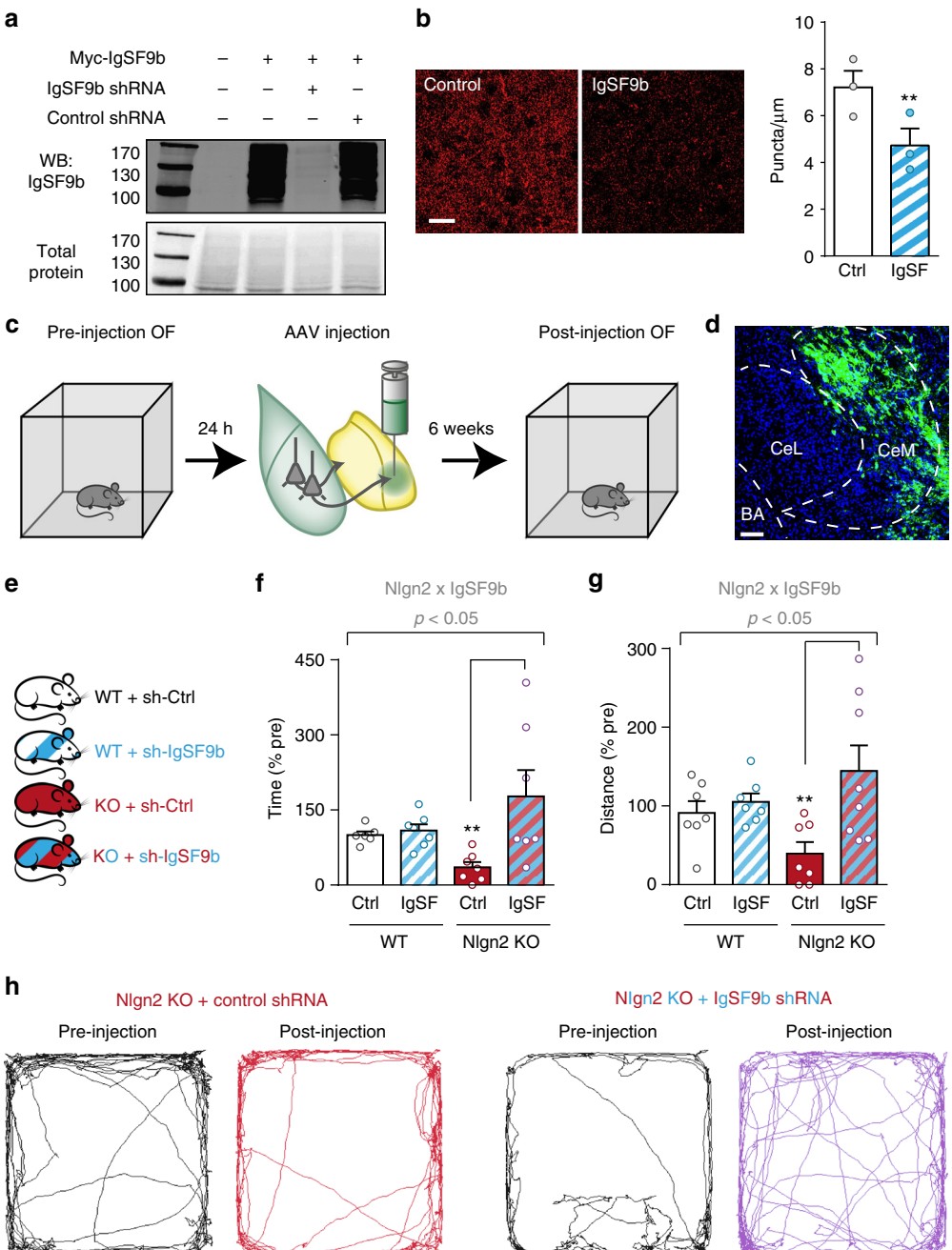

**Fig. 4** Local knockdown of IgSF9b in the adult CeM ameliorates anxiety-related behaviors in Nlgn2 KO mice. **a** Immunoblot for IgSF9b in HEK cell lysates transfected with a Myc-IgSF9b construct and IgSF9b shRNA or control shRNA[14]. Numbers next to the protein ladder represent molecular weight in kDa. **b** Representative images and quantification of perisomatic IgSF9b puncta in the CeM following injection of control shRNA or IgSF9b shRNA. $n = 3$ mice. Scale bar, 20 μm. **c** Schematic diagram showing experimental design of IgSF9b shRNA experiment. Mice were tested in the OF 24 h before (pre) and 6 weeks after (post) injection. **d** Representative image of GFP-positive neurons in viral injection site. **e** Four experimental groups were generated: WT + control shRNA (white), WT + IgSF9b shRNA (white and blue striped), Nlgn2 KO + control shRNA (red), and Nlgn2 KO + IgSF9b shRNA (red and blue striped). **f** Time in center post-injection expressed as % pre-injection. **g** Normalized distance in center post-injection expressed as % pre-injection. **h** Representative tracks of OF exploration of Nlgn2 KO mice pre- and post-injection. Statistically significant ANOVA comparisons are marked in gray at the top of the panels and are listed in Table 1. For all other ANOVA comparisons $F < 1$. Post-hoc analysis: *$p < 0.05$ relative to WT, **$p < 0.01$ relative to WT. Error bars represent SEM, $n = 6$–9 mice per group

respectively), consistent with the increase in mIPSC frequency, indicating that IgSF9b deletion increases the number and/or VIAAT content of inhibitory presynaptic terminals in the CeM. Gephyrin and S-SCAM remained unchanged (Fig. 6d, e, blue bars). IgSF9b deletion also resulted in a trend towards an increase in GABA$_A$R γ2 subunit puncta, as well as a significant reduction

in GABA$_A$R α1 subunit puncta (Fig. 6f, g, blue bars). Nlgn2 KO mice showed a slight reduction in the number of GABA$_A$R α1 subunit puncta, with no effects on any other synaptic marker (Fig. 6c–g, red bars), confirming the observation from the mIPSC recordings that Nlgn2 deletion has only very subtle effects in the CeM.

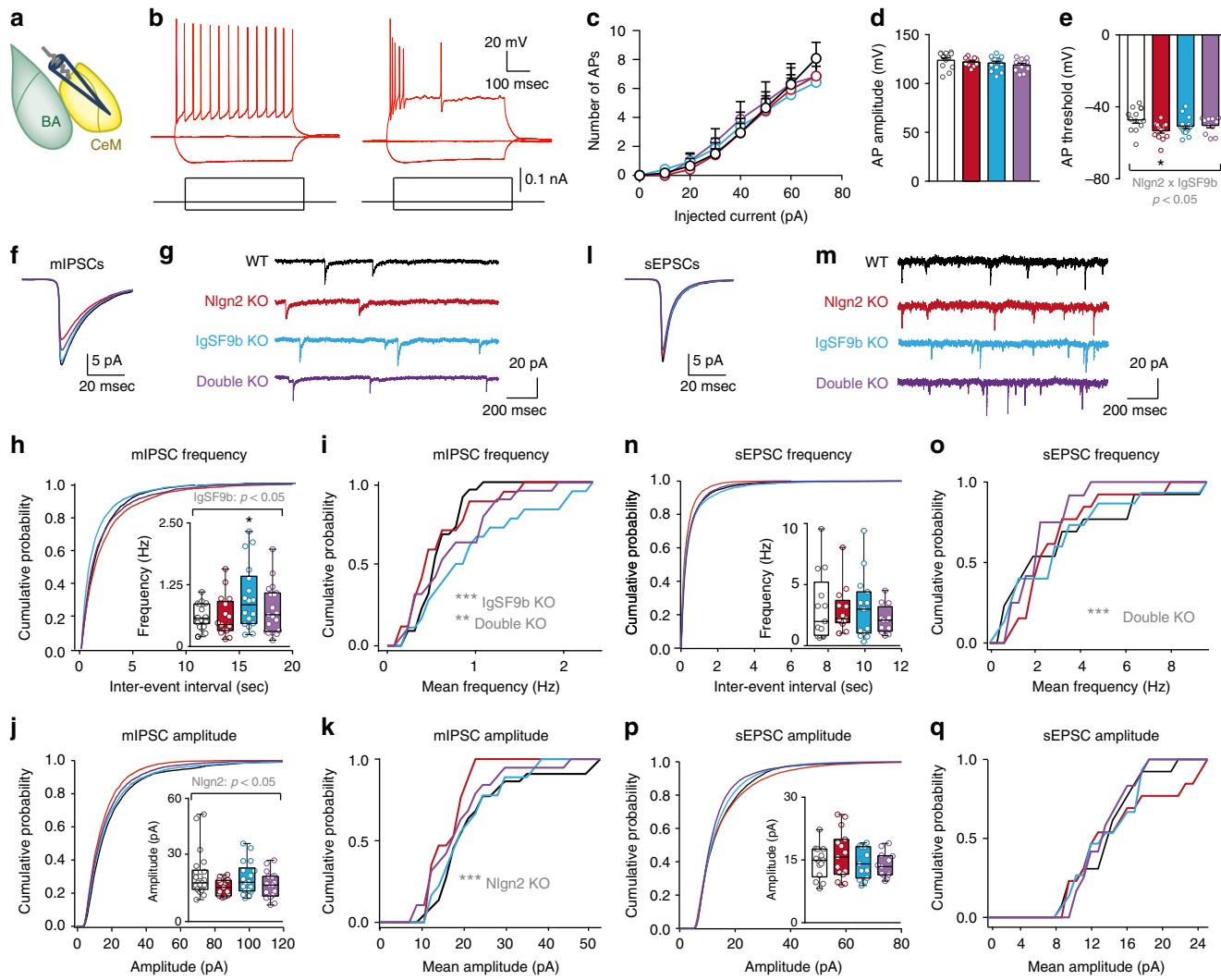

**Fig. 5** IgSF9b deletion upregulates inhibitory synapse function in the CeM. **a** Schematic diagram showing location of mIPSC and sEPSC recordings in the CeM. **b** Representative firing patterns of CeM neurons. **c**–**e** Mean number of action potentials (APs) in response to a 500 ms current step injection (**c**), mean amplitude of the first AP (**d**), and mean AP firing threshold (**e**) in CeM neurons of WT, Nlgn2 KO, IgSF9b KO, and double KO mice. **f**, **g** Mean mIPSCs and representative mIPSC traces. **h** Average cumulative distribution of mIPSC inter-event intervals and quantification of mean mIPSC frequency in the CeM. **i** Probability distribution of mean mIPSC frequencies among all analyzed cells in the CeM. Kolmogorov–Smirnov test: WT vs. IgSF9b KO, $p <$ 0.0001; WT vs. Double KO, $p < 0.001$; Nlgn2 KO vs. IgSF9b, $p < 0.0001$; IgSF9b KO vs. Double KO, $p < 0.01$. **j** Average cumulative distribution of mIPSC amplitudes and quantification of mean mIPSC amplitude in the CeM. **k** Probability distribution of mean mIPSC amplitude of each analyzed cell in the CeM. Kolmogorov–Smirnov test: WT vs. Nlgn2 KO, $p < 0.001$. $n = 17$–22 cells/6–7 mice per genotype. **l**, **m** Mean sEPSCs and representative sEPSC traces. **n** Average cumulative distribution of sEPSC inter-event intervals and quantification of mean sEPSC frequency in the CeM. **o** Probability distribution of mean sEPSC frequencies among all analyzed cells in the CeM. Kolmogorov–Smirnov test: WT vs. Double KO, $p < 0.001$; IgSF9b KO vs. Double KO, $p < 0.001$. **p** Average cumulative distribution of sEPSC amplitudes and quantification of mean sEPSC amplitude in the CeM. **q** Probability distribution of mean sEPSC amplitude of each analyzed cell in the CeM. $n = 13$–15 cells/3–5 mice per genotype. Statistically significant ANOVA comparisons are marked in gray at the top of the panels and are listed in Table 1. For all other ANOVA comparisons, $F < 1$. Post-hoc analysis: *$p < 0.05$ relative to WT, **$p < 0.01$ relative to WT. Error bars represent SEM. Box plots represent median and 25th and 75th percentiles and whiskers are drawn from the minimum to the maximum value

To test whether the increase in VIAAT puncta in the CeM reflects an acute function of IgSF9b on inhibitory synapses or whether this is a developmental consequence, we quantified VIAAT puncta following injection of IgSF9b shRNA into the CeM (Fig. 6h–j). Deletion of IgSF9b increased the number of VIAAT puncta in Nlgn2 KO mice (Fig. 6i, red and blue striped bars), an effect that may underlie the rescue of anxiety-related behavior in Nlgn2 KO mice following local deletion of IgSF9b in the CeM (Fig. 4). Moreover, acute deletion of IgSF9b increased the size of VIAAT puncta in WT mice (Fig. 6j, white and white and blue bars), consistent with the upregulation of VIAAT puncta in single IgSF9b KO mice (Fig. 6c, blue bar). Together, our data indicate

that deletion of IgSF9b in the CeM results in an increase in inhibitory synapse function, which may underlie its behavioral consequences in the anxiety circuitry.

**IgSF9b deletion does not affect BA inhibitory synapses.** Finally, we investigated the consequences of IgSF9b deletion on synaptic inhibition in the BA, in order to determine whether the differential effects of IgSF9b deletion on anxiety-related neuronal activation in the BA and the CeM are also reflected at the synaptic level. We first recorded mIPSCs in excitatory (in our case Venus-negative) neurons in the BA (Fig. 7a–f). Consistent with previous reports, Nlgn2 deletion resulted in a pronounced reduction in

**Table 2 Kinetics of PSCs and passive membrane properties of inhibitory neurons in CeM**

| | WT | Nlgn2 KO | IgSF9b KO | Double KO | F-value[a] | p-value |
|---|---|---|---|---|---|---|
| mIPSCs: rise time (μs) | 830 ± 80 (n = 22) | 916 ± 86 (n = 16) | 855 ± 79 (n = 18) | 781 ± 51 (n = 18) | Interaction: $F_{1,70} = 1.10$ | 0.30 |
| mIPSCs: decay (ms) | 14.97 ± 0.71 (n = 22) | 17.17 ± 0.89 (n = 17) | 15.76 ± 1.16 (n = 18) | 15.25 ± 0.89 (n = 19) | Interaction: $F_{1,70} = 2.19$ | 0.14 |
| sEPSCs: rise time (μs) | 457 ± 39 (n = 13) | 400 ± 35 (n = 15) | 515 ± 63 (n = 13) | 426 ± 44 (n = 15) | Nlgn2 KO: $F_{1,52} = 2.36$ | 0.13 |
| sEPSCs: decay (ms) | 3.13 ± 0.17 (n = 13) | 2.90 ± 0.15 (n = 15) | 3.07 ± 0.19 (n = 15) | 2.80 ± 0.16 (n = 13) | Nlgn2 KO: $F_{1,52} = 2.10$ | 0.15 |
| Membrane resistance (MOhm) | 477 ± 40 (n = 32) | 501 ± 44 (n = 30) | 489 ± 38 (n = 32) | 397 ± 31 (n = 28) | IgSF9b KO: $F_{1,115} = 1.62$ Interaction: $F_{1,115} = 2.56$ | 0.21,0.11 |
| Membrane capacitance (pF) | 28.86 ± 1.42 (n = 33) | 31.75 ± 1.92 (n = 31) | 28.75 ± 1.76 (n = 33) | 32.71 ± 2.07 (n = 28) | Interaction: $F_{1,103} = 3.91$ | 0.05 |

[a]For all other ANOVA comparisons, $F < 1$

mean frequency and mean amplitude of mIPSCs in the BA (Fig. 7c, e, red bars)[8,20,34,35]. In striking contrast, IgSF9b deletion affected neither mean mIPSC frequency nor mean IPSC amplitude (Fig. 7c, e, blue bars). Double KO mice showed a trend towards a reduction in mean frequency and mean amplitude of IPSCs that was similar to that of Nlgn2 KO mice (Fig. 7c, e, purple bars). Comparison of the probability distribution of mean frequencies or mean amplitudes among all cells to the corresponding WT distribution (Fig. 7d, f) revealed a significant shift towards lower mean frequencies in double KO cells (Fig. 7d, black vs. purple line), indicating that individual subpopulations of cells may be differentially affected in double KO mice. Consistent with the decrease in mIPSC frequency and amplitude, Nlgn2 deletion significantly reduced the number and size of perisomatic gephyrin and GABA$_A$R α1 puncta, while no changes were observed for VIAAT or, interestingly, the GABA$_A$R γ2 subunit (Fig. 7g–j, red bars). IgSF9b deletion did not affect any inhibitory synapse markers in the BA, while double KO mice showed reductions in gephyrin and GABA$_A$R α1 staining that were identical to those observed in Nlgn2 KO mice (Fig. 7g–j, blue and purple bars, respectively), consistent with the notion that IgSF9b does not normalize anxiety through a synaptic mechanism in the BA. Thus, combined with the functional changes observed in whole-cell recordings in amygdala slices, these morphological data indicate that IgSF9b and Nlgn2 differentially affect inhibitory synapses in the amygdala, with Nlgn2 deletion primarily impairing synaptic inhibition in the BA, and IgSF9b deletion primarily enhancing synaptic inhibition in the CeM. Altogether, our findings are consistent with a model in which IgSF9b deletion normalizes anxiety-related behaviors and neuronal activity by increasing inhibition onto CeM output neurons and hence counteracting the anxiety-related overactivation of the CeM (Fig. 8).

## Discussion

In the present study we sought to elucidate the role of the cell adhesion molecule IgSF9b and its interactions with Nlgn2 in amygdala circuits, synapses, and behaviors related to anxiety processing. We show that deletion of IgSF9b has anxiolytic consequences and normalizes the prominent anxiety phenotype observed in Nlgn2 KO mice. This normalization does not occur through a mechanistic interaction of Nlgn2 and IgSF9b at the same synapses, but likely through differential effects on different inhibitory synapses in the BA and CeM, respectively. Specifically, our data support a model in which reduced inhibition in BA of Nlgn2 KO mice results in overactivation of BA → CeM projection neurons under anxiogenic conditions, which is counteracted in the CeM by the increased inhibition resulting from additional deletion of IgSF9b (Fig. 8). Together, our data provide the first description of IgSF9b function in vivo and uncover a novel role for IgSF9b in anxiety-related behavior and amygdala inhibitory synapses. Moreover, our findings highlight that IgSF9b-expressing synapses and neurons in the CeM may represent an important common target for anxiolytic treatments that is independent of individual upstream mutations.

Major efforts have recently been invested in determining the full complement of proteins that governs the formation and function of inhibitory synapses[12,15,36–40]. The cell adhesion molecule IgSF9b was identified as one of these candidate molecules in cell cultures[14,15], but the mechanism by which it regulates inhibitory synapse function in vivo in intact circuits remained completely unknown. Here we show that in the CeM, deletion of IgSF9b results in an enhancement of inhibitory synaptic transmission, likely through a presynaptic mechanism. In particular, the increases in mIPSC frequency and VIAAT staining without a concomitant increase in gephyrin indicate that IgSF9b deletion may result in an increase in the number of VIAAT-positive vesicles per synaptic terminal, analogous to effects recently observed in IgSF21 KO mice (albeit with opposite polarity)[41]. Together with the observation that IgSF9b, like its mammalian paralog IgSF9 and its Drosophila ortholog Turtle, forms primarily or exclusively homophilic interactions[14,16,42], our data indicate that IgSF9b provides a transsynaptic signaling complex that regulates the organization of the inhibitory presynaptic terminal. Moreover, the differential effects of IgSF9b deletion on GABA$_A$R α1 and γ2 subunits in the CeM indicate that IgSF9b may also regulate the subunit composition of GABA$_A$Rs at inhibitory postsynapses, although the mechanism or functional significance of this effect remains unknown.

Intriguingly, IgSF9b appears to play distinct roles at different inhibitory synapses, as evidenced by our finding that IgSF9b deletion affects inhibitory synaptic transmission and synaptic markers only in the CeM but not the BA, despite being expressed in both (Supplementary Fig. 6). One possible explanation for this difference lies in the differential composition of the two structures with respect to neuron types[9,43]. The BA is a cortex-like structure composed primarily of excitatory projection neurons with a small number of inhibitory interneurons, while the CeA is similar to striatum in that it is composed almost exclusively of inhibitory neurons. Therefore, the vast majority of inhibitory synapses sampled in the BA and CeM are made onto excitatory and inhibitory neurons, respectively, raising the possibility that IgSF9b differentially affects these synapse subtypes. Consistent

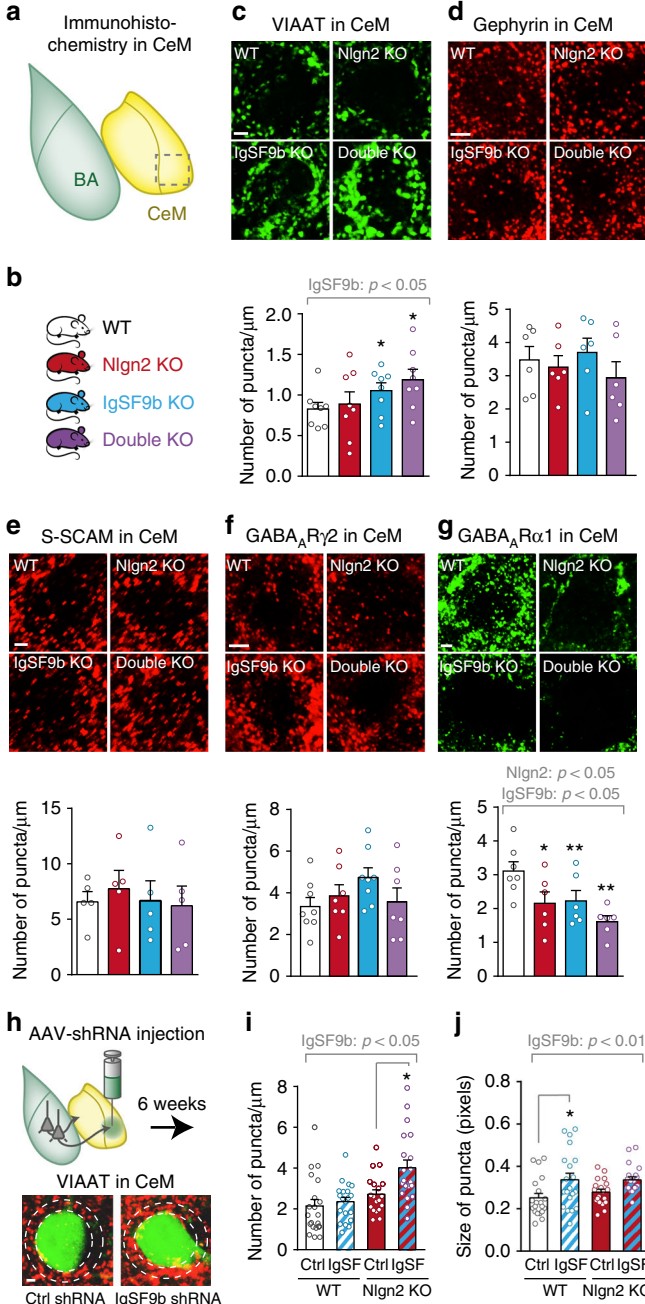

with this notion, IgSF9b was previously proposed to function primarily at inhibitory synapses onto interneurons in hippocampal cultures[14]. In contrast, there is growing evidence that other known synapse organizers such as Nlgn2, MDGA1, and GARLH/LHFPL4 function exclusively at inhibitory synapses onto excitatory neurons[8,44,45]. Together, these findings give rise to the notion that excitatory and inhibitory neurons may utilize an entirely different complement of organizer proteins at inhibitory postsynapses, and that IgSF9b may be the first example of a synapse organizer with specificity for inhibitory synapses onto inhibitory neurons.

Moreover, the function of IgSF9b at inhibitory synapses may also depend on the identity of the presynaptic neuron. In hippocampal cultures, shRNA-mediated deletion of IgSF9b resulted in a decrease in mIPSC frequency and a reduction in VIAAT-positive gephyrin clusters[14], in contrast to the increase in mIPSC frequency and VIAAT puncta observed here in the CeM (Figs. 5, 6). Since the inhibitory neuron subtypes found in the CeM are entirely distinct from those observed in the hippocampus[10,11], it is conceivable that these different subtypes may differentially employ IgSF9b at inhibitory postsynapses. Alternatively, IgSF9b may play distinct roles at different stages of synaptic development (i.e., at developing synapses in neuronal cultures vs. mature synapses in adult mice) or in dissociated cultures vs. intact neuronal networks. Cell type- and circuit-specific analysis of IgSF9b function will be essential for fully understanding its role in the brain.

Given that IgSF9b and Nlgn2 function at distinct synapses and do not appear to interact in a cell-autonomous manner, at least in the amygdala, how does deletion of IgSF9b normalize the prominent anxiety phenotype observed in Nlgn2 KO mice? Combined evidence from our cFos analysis (Fig. 2a), retrograde tracing experiments (Fig., 2c–e), and in vivo electrophysiology (Fig. 3) indicates that IgSF9b deletion normalizes anxiety-related output specifically in the CeM of Nlgn2 x IgSF9b double KO mice. Given that the normalization of anxiety-related behaviors can be mimicked by local shRNA-mediated knockdown of IgSF9b in the CeM of Nlgn2 KO mice (Fig. 4), it is likely to occur through a local mechanism within the CeM. In light of our observation that IgSF9b deletion increases inhibitory synaptic transmission in the CeM (Figs. 5, 6), the most parsimonious explanation is that this increased inhibition onto CeM output neurons counteracts the increased excitation originating from Nlgn2 KO BA projection neurons, thus balancing the activity of CeM neurons mediating anxiogenic projections to the brainstem[9,10] (Fig. 8). This model may also explain why the anxiolytic effect of IgSF9b in WT mice is relatively modest: in the absence of a pathological overactivation of CeM projection neurons such as that induced by Nlgn2 deletion, the increased inhibition may have relatively minor effects on WT CeM output.

An interesting remaining question regards the source of the inhibitory inputs to the CeM that are upregulated by IgSF9b deletion. CeM neurons are known to receive feedforward inhibitory projections from the CeL, the intercalated nucleus and the bed nucleus of the stria terminalis[9,10], and synaptic transmission at any of these inhibitory connections or at local interneurons may be upregulated in response to IgSF9b deletion. Importantly, our model does not require the neuronal activity of the upstream inhibitory neurons to be increased, and indeed we show that CeL inputs to the CeM are not differentially activated (Fig. 2e). We cannot rule out that other CeM inputs may show altered activation in IgSF9b or double KO mice. However, the effectiveness of the local deletion of IgSF9b in CeM in mimicking the double KO phenotype indicates that any such upstream alterations cannot be required for the normalization of anxiety-related behaviors by IgSF9b deletion. Our findings, therefore, identify local information processing in the CeM as a key

**Fig. 6** IgSF9b deletion increases VIAAT puncta in the CeM. **a**, **b** Schematic diagram showing location of immunohistochemistry analysis in the CeM (**a**) and genotypes assessed (**b**). **c–g** Photomicrographs and quantification of the number of perisomatic puncta of **c** VIAAT, **d** gephyrin, **e** S-SCAM, **f** GABA$_A$Rγ2 and **g** GABA$_A$Rα1 in all four genotypes. Scale bar, 2 μm. Error bars represent SEM, $n = 5$–8 mice per genotype. **h** Schematic diagram showing experimental design and representative photomicrographs of VIAAT puncta surrounding GFP-positive neurons in the CeM of Nlgn2 KO mice injected with control shRNA (left image) or IgSF9b shRNA (right image). Scale bar, 2 μm. **i**, **j** Number (**i**) and size (**j**) of VIAAT puncta in WT + control shRNA (white), WT + IgSF9b shRNA (white and blue striped), Nlgn2 KO + control shRNA (red), Nlgn2 KO + IgSF9b shRNA (red and blue striped). $n = 20$ cells/3 mice per experimental group. Statistically significant ANOVA comparisons are marked in gray at the top of the panels and are listed in Table 1. For all other ANOVA comparisons $F < 1$. Post-hoc analysis: *$p < 0.05$ relative to WT, **$p < 0.01$ relative to WT, ***$p < 0.001$ relative to WT. Error bars represent SEM. WT, white bars; Nlgn2 KO, red bars; IgSF9b KO, blue bars; double KO, purple bars

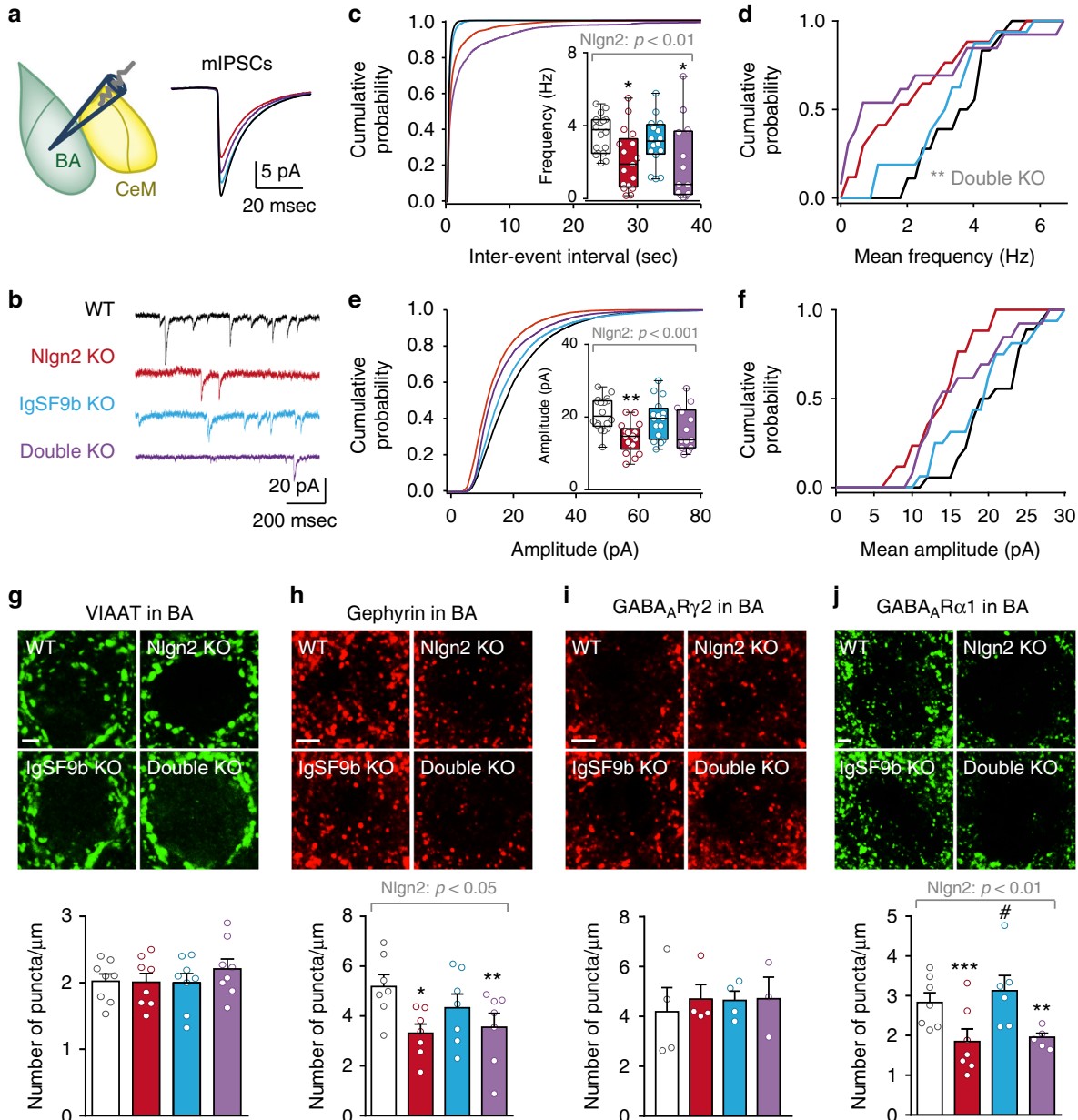

**Fig. 7** IgSF9b deletion does not affect inhibitory synapses in the BA. **a**, **b** Schematic diagram illustrating recording sites in the BA, mean mIPSCs and representative mIPSC traces from the BA of WT, Nlgn2 KO, IgSF9b KO, and double KO mice. **c** Average cumulative distribution of mIPSC inter-event intervals and quantification of mean mIPSC frequency in the BA. **d** Probability distribution of mean mIPSC frequency of each analyzed cell in the BA. Kolmogorov–Smirnov test: WT vs. Double KO, $p < 0.0001$. **e** Average cumulative distribution of mIPSC amplitudes and quantification of mean mIPSC amplitude in the BA. **f** Probability distribution of mean mIPSC amplitude among all analyzed cells in the BA. $n = 13$–18 cells/5–6 mice per genotype. **g**–**j** Photomicrographs and quantification of the number of perisomatic puncta of **g** VIAAT, **h** gephyrin, **i** GABA$_A$R$\gamma$2, and **j** GABA$_A$R$\alpha$1 in all four genotypes. Scale bar, 2 μm. $n = 3$–8 per genotype. Statistically significant ANOVA comparisons are marked in gray at the top of the panels and are listed in Table 1. For all other ANOVA comparisons $F < 1$. Post-hoc analysis: $*p < 0.05$ relative to WT, $**p < 0.01$ relative to WT, $***p < 0.001$ relative to WT, $\#p < 0.05$ relative to double KO. Error bars represent SEM. Box plots represent median and 25th and 75th percentiles and whiskers are drawn from the minimum to the maximum value. WT, white bars; Nlgn2 KO, red bars; IgSF9b KO, blue bars; double KO, purple bars

mediator of anxiety-related behaviors and highlight the fundamental importance of better understanding the role of the CeM in anxiety processing.

The strong correlation of power of beta oscillations with distance from the center of the OF in Nlgn2 KO mice and its exacerbated increase during risk-assessment behavior implicates beta oscillations in the CeM as a key neural signature of pathological anxiety. To our knowledge this is the first time that oscillatory activity in the CeM is described in anxiety processing in general,

and beta oscillations during risk-assessment behavior in particular. At present, both the function of these beta oscillations in anxiety processing and the mechanisms that underlie their modulation by Nlgn2 and IgSF9b remain unknown. Beta oscillations have previously been observed in the basal ganglia and somatosensory cortex during decision-making tasks[46,47], in the hippocampus upon exposure to novelty, and in the mediodorsal thalamus during working memory-related tasks[48,49], highlighting their multifaceted role in information processing. However, their

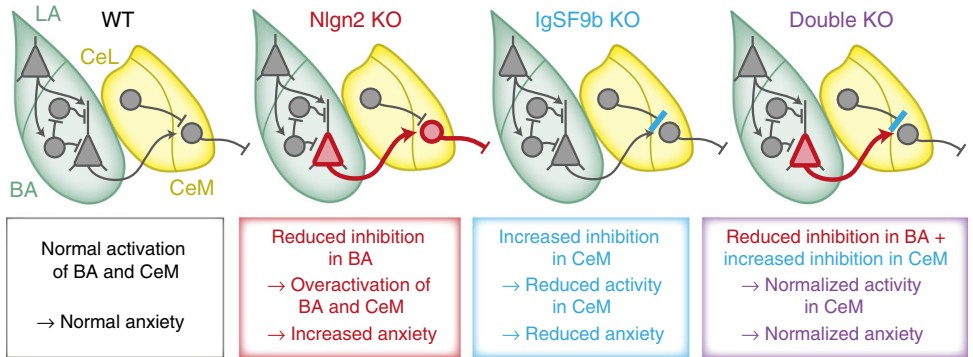

**Fig. 8** Model of the circuitry-based normalization of the anxiety phenotype in Nlgn2 x IgSF9b double KO mice. Based on our findings, we propose the following model for the normalization of anxiety-related behaviors in Nlgn2 x IgSF9b double KO mice: Nlgn2 deletion enhances activity of excitatory BA → CeM projection neurons, resulting in overactivation of anxiogenic projection neurons in the CeM and hence increased anxiety-related behaviors. IgSF9b deletion enhances synaptic inhibition onto anxiogenic projection neurons in the CeM, which results in reduced activation of CeM projection neurons and hence reduced anxiety-related behaviors. In Nlgn2 x IgSF9b double KO mice, the combination of these two effects results in normalization of the activity of CeM projection neurons and hence a normalization of anxiety-related behaviors

precise function in mediating these behavioral processes remains controversial and an area of active investigation. Similarly, the cellular and synaptic mechanisms leading to the generation of beta rhythms are largely unknown, although changes in inhibitory synaptic activity are known to affect LFPs in general[50] and beta activity in specific[51]. Further experiments will be essential to fully understand how Nlgn2 and IgSF9b regulate beta oscillations in the CeM and how this contributes to the generation and normalization of pathological anxiety processing.

The anxiolytic effects of IgSF9b deletion in Nlgn2 KO mice are intriguing in light of previous studies showing that variants in IgSF9b are associated with major depression and the negative (mainly affective) symptoms of schizophrenia[17–19]. It is conceivable that the IgSF9b variants in human patients are in fact gain-of-function rather than loss-of-function mutations, which based on our findings, could result in reduced synaptic inhibition in the CeM. Indeed, our data indicate that the function of IgSF9b-containing synapses in the CeM may be key in determining the vulnerability or resilience of individuals towards upstream anxiogenic factors. The precise molecular mechanisms remain to be elucidated both in mice and humans, but it is clear that IgSF9b is emerging as an important regulator of affective behaviors, and that further investigations into its function may substantially contribute to our understanding of a range of psychiatric disorders.

Ultimately, perhaps the most exciting conclusion arising from our study is that IgSF9b-expressing neurons and synapses in CeM may represent a viable common target for anxiolytic therapies, independent of the upstream anxiogenic mutations. Given that both global and local deletion of IgSF9b lead to a remarkably specific anxiolytic effect, targeting IgSF9b pharmacologically may provide a promising strategy for the development of more selective anxiolytic therapies. Moreover, it is tempting to speculate that in the age of circuit psychiatry[26], targeting IgSF9b-expressing neurons with viral vectors will become feasible, offering entirely new treatment options for patients with anxiety and co-morbid psychiatric disorders.

## Methods

**Experimental subjects**. Nlgn2 KO mice[21] were generated in our laboratory on an 129/Sv background and were backcrossed onto a C57BL/6 J background for at least six generations. IgSF9b KO mice were obtained from Lexicon Pharmaceuticals (The Woodlands, TX, U.S.A.; Omnibank clone 281214, generated through insertion of the Omnibank gene trap vector 48 into the IgSF9b gene in Sv129 ES cells) and were backcrossed onto a C57BL/6 J background for at least six generations. Nlgn2 KO mice and IgSF9b KO mice were crossed to generate Nlgn2 het x IgSF9b

het breeding pairs, from which four experimental genotypes were obtained as littermates, i.e., WT, Nlgn2 KO, IgSF9b KO, and double KO mice. For in vitro electrophysiology experiments, these mice were additionally crossed with mice expressing a Venus transgene under the control of the VIAAT promoter to label inhibitory interneurons (generated using a Venus construct generously provided by Dr. Atsushi Miyawaki, RIKEN)[32]. All mice were 2–3 months old at the beginning of the experiment. For experiments involving OF testing (Figs. 1–4 and Supplementary Fig. 1–4), only male mice were used unless specified otherwise. For the EPM experiment (Fig. 1h–m) and for molecular and slice electrophysiology experiments (Figs. 5–7), both male and female mice were used, based on the observation that the OF phenotype is identical in both sexes. Animals were maintained on a 12 h light/dark cycle (7 am/7 pm), with food and water ad libitum. All experiments were performed during the light cycle (with the exception of home cage activity monitoring as described below, Supplementary Fig. 2). The experimenter was blind to genotype during all stages of data acquisition and analysis. All procedures were approved by the State of Niedersachsen (Landesamt für Verbraucherschutz und Lebensmittelsicherheit) and were carried out in agreement with the guidelines for the welfare of experimental animals issued by the Federal Government of Germany and the Max Planck Society.

**Behavioral characterization**. The OF test was conducted in a square arena made of white plastic (50 cm × 50 cm). Mice were placed in the corner of the OF and were permitted to explore the arena for 10 min. The EPM was conducted in an apparatus made of gray plastic, with two open arms and two closed arms (28 cm × 5 cm each, elevated 45 cm from the ground). Mice were placed in the closed arm and were permitted to explore the apparatus for 5 min. Performance was recorded using an overhead camera system and scored automatically using the Viewer software (Biobserve, St. Augustin, Germany). Between each mouse, the arena was cleaned thoroughly with 70% ethanol followed by water to eliminate any odors left by the previous mouse.

In our previous study[8], the center zone of the OF was defined as a square of 25 × 25 cm. However, closer assessment revealed that a second, intermediate zone (12.5 cm–25 cm from the walls of the chamber) was also avoided by the Nlgn2 KO mice (Supplementary Fig. 1a-h), and that in fact the behavior in this intermediate zone resembled the behavior in the center, rather than the periphery, in all four genotypes assessed in the present study. For this reason, an extended center zone of 37.5 × 37.5 cm was used throughout this study to more accurately reflect the anxiogenic area.

Recording of home cage activity was performed using the LABORAS system and software (Metris, Hoofddorp, The Netherlands). Mice were habituated to the LABORAS cages for two days. On the third day, activity was recorded for two 6-hour periods, from 9 pm to 3 am (dark cycle) and from 9 am to 3 pm (light cycle). The following parameters were assessed: total duration of locomotor activity, immobility, grooming, and climbing, as well total distance traveled and average velocity during locomotion.

**cFos induction assay**. To assess anxiety-induced cFos activation, mice were first exposed to the OF arena for 10 min. Ninety minutes after exposure, they were anesthetized with Avertin (Tribromoethanol, Sigma) and perfused transcardially first with saline, then with 4% paraformaldehyde (PFA) in 0.1 M phosphate buffer (PB). Brains were post-fixed in PFA overnight, and cryoprotected in 30% sucrose in 0.1 M PB. Free-floating sections (40 μm thickness) were prepared using a Leica CM3050S cryostat (Leica, Wetzlar, Germany). Sections were incubated in blocking

solution (3% bovine serum albumin, 10% goat serum, 3% Triton X-100 in 0.1 M phosphate-buffered saline (PBS)) for 1 h, then incubated for 12 h with Rabbit polyclonal anti-cFos antibody (catalog# sc-52, Santa Cruz Biotech, Dallas, Texas, USA) diluted 1:2000 in blocking solution; and then incubated for 2 h with Alexa Fluor 488 goat anti-rabbit antibody (Invitrogen, Eugene, OR, USA) diluted 1:600 in blocking solution. The sections were washed with (PBS) after each incubation, and were finally mounted on glass slides using Aqua-Poly/Mount (Polysciences, Eppelheim, Germany).

For PV/cFos or SOM/cFos immunolabeling, amygdala sections were processed as described above for cFos immunohistochemistry. All secondary antibodies were diluted 1:600 in respective blocking solution and obtained from Invitrogen, Eugene, USA. For PV immunolabeling, sections were incubated for 12 h with mouse monoclonal anti-PV antibody (catalog# 235, SWANT, Bellinzona, Switzerland) diluted 1:2000 in blocking solution; and then incubated for two hours with Alexa Fluor 555 goat anti-mouse antibody. For SOM immunolabeling, the blocking solution contained 10% donkey serum and 0.3% Triton X-100 in PBS 0.1 M. Sections were incubated for 12 h in goat polyclonal anti-SOM antibody (catalog# sc-7819, Santa Cruz Biotech, Dallas, Texas, USA) diluted 1:1000 in blocking solution; and then with Alexa Fluor 488 donkey anti-goat antibody. Sections were washed with PBS after each incubation, and were finally mounted on glass slides using Aqua-Poly/Mount (Polysciences, Eppelheim, Germany).

Images of cellular and cellular markers were obtained using a confocal laser scanning microscope (Leica SP2) with a ×40 oil immersion objective. For each set, sections were anatomically matched and the settings for laser power, gain and offset were kept constant during imaging. 2 stacks of 5 μm thickness and containing 2 optical sections each were obtained from each amygdala section. In total, 10–12 stacks were imaged from 5–6 amygdala sections per mouse in each group. cFos images were thresholded manually in ImageJ with threshold set as 3*background intensity and the same threshold value was applied for WT, Nlgn2 KO, IgSF9b KO, and double KO in each group. Single-labeled cells were quantified using the Create Spots tool in Imaris (Bitplane, Zurich, Switzerland). To obtain the number of double- labeled cells, the Colocalize Spots tool in Imaris was used.

**Retrograde labeling**. Mice received an intraperitoneal (i.p.) injection of Carprofen (5 mg/kg) to reduce post-surgery pain 30 min prior to surgery. Mice were anaesthetized with Avertin by i.p. injection (20 ml/kg body weight) and placed in digital stereotaxic frame. To bilaterally label BA neurons projecting to the CeM, 50 nl of red Retrobeads (excitation maximum at 530 nm and emission maximum at 590 nm) were injected into the CeM (0.70 mm posterior, ± 2.35 mm medial, and 5.08 mm ventral from bregma). A Hamilton syringe (1 μl) was used to manually deliver the Retrobeads at the rate of 0.5 nl/sec. After the injection was completed, the tip of the syringe was raised by 100 μm and left for 3 min to allow diffusion of the Retrobeads at the injection site; and then slowly withdrawn at the rate of 1 mm/min. Following surgery, mice received Metamizol with drinking water (200 mg/kg/day, drinking rate estimated at 3 ml/day) for 3 days to reduce pain and risk of inflammation.

Mice were single housed for 7 days post-surgery to allow their full recovery and traveling of beads from the injection site up the axons of BA neurons to their somata. To induce anxiety-associated neuronal activation, mice were subjected to the OF for 10 min, and cFos was subsequently assessed as described above. Only slices from brains in which injection sites did not exceed the borders of CeM (confirmed by visualizing the sites of injection on 5 subsequent coronal sections of amygdala spanning 400 μm of tissue) were included in subsequent imaging and analysis. Data acquisition and quantification of cFos-positive cell bodies containing Retrobeads were performed similarly to quantification of double labeled neurons as described above. Sections were also used for histological verification of the Retrobead injection site, and mice were excluded from analysis if the injection site lay outside the CeM (Supplementary Fig. 3j).

**In vivo electrophysiology**. Mice were anesthetized with Avertin (loading dose 20 ml/kg, maintenance dose 2 ml/kg i.p.) and placed in a stereotaxic frame, and their body temperature was monitored by a rectal probe and maintained at 36 °C. An incision in the midline of the scalp was made to expose the skull. Bregma and lambda were aligned to a plane level ± 50 μm. A multi-wire electrode array was unilaterally implanted targeting the left CeM (0.9 mm posterior, 2.3 mm lateral and 5.04 mm ventral to bregma). The electrodes consisted of 2 bundles (spaced 750–950 μm) of 8 individual insulated tungsten wires (13 μm inner diameter, impedance 60–100 kΩ) inserted into a polymide tube (127 μm inner diameter) and attached to an 18-pin connector. A reference screw was implanted above the cerebellum. The implant was secured with two screws implanted in the skull ~300 μm lateral and anterior to the electrode and bonded with dental cement. Immediately after the surgery, mice subcutaneously received an analgesic (Carprofen 5 mg/kg) and an antibiotic (Baytril 5 mg/kg). Twenty four hours after the surgery, mice received Carprofen (5 mg/kg) subcutaneously and Baytril in the drinking water (0.2 mg/ml). Mice were sacrificed following the OF recording for histological verification of the recording site (Supplementary Fig. 4). Mice in which the electrode was not implanted in the CeM were excluded from the analysis.

Male mice were single housed for 7 days after the surgery and before the recording. For data acquisition, the mice were connected to the electrophysiological equipment and placed in the OF chamber, where they were allowed to explore for

15 min. The electrophysiological signal was amplified and sent to the acquisition board. The raw signal was acquired at 32 kHz sampling rate, band pass filtered (0.1–9000 Hz), and stored for offline analysis. During the experiment, simultaneous electrophysiological and video recordings were made by the Cheetah Data Acquisition System.

LFPs were analyzed using custom-written MATLAB scripts. The signal was filtered between 0.7 and 400 Hz using a zero-phase distortion FIR filter and down sampled to 1 kHz. The multitaper method was used for the power analysis (Chronux Package)[52]. The following time windows were used: theta range (4–12 Hz), 1 s with 0.8 s of overlap; beta range (18–30 Hz), 1 s with 0.5 s of overlap; gamma range, 0.15 s with 0.1 s of overlap. To calculate the power spectra during the entire OF session, 5 tapers were used with a time bandwidth of 3. For each mouse, the time periods and the tracks of the movement in the periphery and center were extracted using a modified version of the autotyping toolbox[53]. To evaluate the relative increase in beta power as a function of distance from the center, the distance from the center during the entire OF session was binned for each mouse (3 cm bins). To compute the beta power, the power was first summed across the beta band, and then the summed values were averaged for each location bin. The individual bin values were normalized by the average power in the periphery ( > 18 cm from center) and the linear correlation coefficients (fitlm function, MATLAB) per genotype were computed. To calculate the correlation between speed and beta power, the beta power was averaged in speed bins of 5 cm/s for each mice and the linear correlation coefficients computed as above.

To evaluate power changes during the SAP, the time events at which the mouse showed a clear SAP from the periphery towards the center of the OF were manually extracted. The events were identified by a typical elongation of the body and a very slow forward movement that was always followed by a retreat movement[30,31]. For the beta band, the power across frequencies was summed to produce one value for each time point. Then, the power values at each time point during the SAP were averaged and the mean power values per event were averaged per group. The Morlet wavelet transform was used to visualize the power at different frequencies ranges as shown in Fig. 3k, with 40 wavelets at centered frequencies ranging from 1 to 120 Hz and a length of 10 cycles.

**Analysis of synaptic markers**. Immunolabeling for VIAAT was performed on perfusion-fixed brains as described for the cFos assay. Briefly, sections were incubated for 24 h with rabbit polyclonal anti-VIAAT antibody (catalog# 131002, Synaptic Systems, Goettingen, Germany) diluted 1:1000 in blocking solution (3% bovine serum albumin, 10% goat serum, 3% Triton X-100 in 0.1 M PBS), washed, and then incubated for 2 h with Alexa Fluor 488 goat anti-rabbit antibody (Invitrogen, Eugene, OR, USA) diluted 1:600 in blocking buffer. Immunolabeling for IgSF9b, Nlgn2, gephyrin, S-SCAM, GABA$_A$Rα1, and GABA$_A$Rγ2 was performed on methanol-fixed fresh frozen brain sections using a modified version of a published protocol[54]. Brains were frozen immediately after dissection in an isopentane bath at −35 °C to −40 °C. Coronal sections were prepared using a Leica CM3050S cryostat (Leica, Wetzlar, Germany), mounted on glass slides, and dried at room temperature. Sections were then fixed in methanol at −20 °C for 5 min, and blocked for 1 h. The incubation was 12 h with primary antibody and 2 h with secondary antibody for each immunolabeling. The following primary antibodies were used, diluted in blocking solution: Rabbit polyclonal anti-IgSF9b (catalog# HPA010802, Sigma Aldrich, Darmstadt, Germany) at 1:1000 mouse monoclonal anti-Nlgn2 (catalog# 129511, Synaptic Systems, Goettingen, Germany) at 1:1000; mouse monoclonal anti-gephyrin (catalog# 147111BT, Synaptic Systems, Goettingen, Germany) at 1:1000; rabbit polyclonal anti-MAGI2 ( = S-SCAM, catalog# M2441, Sigma Aldrich, Darmstadt, Germany); rabbit polyclonal anti-GABA$_A$Rα1 (catalog# 224203, Synaptic Systems, Goettingen, Germany) at 1:1000; guinea pig polyclonal GABA$_A$Rγ2 (generously provided by Dr. Jean-Marc Fritschy, University of Zürich) at 1:1000. The following secondary antibodies were obtained from Invitrogen, Eugene, USA, and diluted 1:600 in the blocking solution: Alexa Fluor 555 goat anti-rabbit antibody, Alexa Fluor 488 goat anti-rabbit antibody, Alexa Fluor 555 goat anti-mouse antibody. Sections were washed with PBS after each incubation. The slides were then dried overnight at 4 °C, and covered with mounting media (Aqua-Poly/Mount; Polysciences, Eppelheim, Germany) and glass coverslips.

Images of synaptic markers were obtained using a confocal laser scanning microscope (Leica SP2) with a ×63 oil immersion objective and ×8 digital zoom. For each set, sections were anatomically matched and the settings for laser power, gain and offset were kept constant during imaging. 12 stacks of 2 μm thickness and containing 2 optical sections each were obtained from each amygdala section (12 stacks from 4 sections for each mouse in total). Images were thresholded in ImageJ, with same threshold applied to all mice in each set. To quantify perisomatic synapses, the perisomatic area was identified by manually tracing the perimeter of the cell body (defined as a circular area largely devoid of immunofluorescence)[6,8]. The perimeter was then expanded by 1.4 μm or 2 μm in each direction for quantification of postsynaptic puncta or presynaptic puncta, respectively. The number of particles was quantified in this area using the "count particles" module in ImageJ, and the number of particles per area was divided by the length of the cell body perimeter to obtain the final result.

**In vitro electrophysiology**. Adult (8–12-week-old) WT, Nlgn2 KO, IgSF9b KO, and double KO mice additionally expressing a VIAAT-Venus transgene[32] were

anesthetized with Avertin and perfused transcardially for 90 s with an ice-cold sucrose-based solution (6 mM MgCl$_2$ 0.1 mM CaCl$_2$, 50 mM sucrose, 2.5 mM glucose, and 3 mM kynurenic acid diluted in artificial cerebrospinal fluid (aCSF, 124 mM NaCl, 2.7 mM KCl, 26 mM NaHCO$_3$, and 1.25 mM NaH$_2$PO$_4$)) as described previously for the preparation of amygdala slices from adult mice[55]. The brains were rapidly dissected and placed in the same ice-cold sucrose-based solution. The brainstem was removed and the brains were mounted on a holder and transferred to the vibratome chamber for preparation of 300 µm coronal sections. Slices containing the BA and central amygdala were transferred to a chamber filled with aCSF (see above) with additional 2 mM CaCl$_2$ and 1.3 mM MgCl$_2$ and equilibrated with 95% O$_2$/5% CO$_2$. Slices were allowed to recover for 20 min at 33 °C and maintained at room temperature thereafter. All chemicals were obtained from Merck Millipore (Molsheim, France), Sigma Aldrich (Darmstadt, Germany), or Tocris Bioscience (Bristol, UK).

Whole-cell patch-clamp recordings were obtained at room temperature (~22 °C) with an EPC10 amplifier (HEKA Elektronik, Germany). Slices were kept in a recording chamber and perfused with aCSF with additional 1.3 mM MgCl$_2$, 2 mM CaCl$_2$, 18.6 mM glucose, and 2.25 mM ascorbic acid (osmolarity ≈320 mOsm) at a rate of 1–2 ml/min. Neurons were visually identified with infrared video microscopy using an upright microscope equipped with a ×60 objective. VIAAT-positive neurons were identified by Venus expression. For recordings in BA, VIAAT-Venus-negative neurons were targeted, while for recordings in CeM, VIAAT-Venus-positive neurons were targeted. Patch electrodes (3–5 MΩ open tip resistance when filled with internal solution) were pulled from borosilicate glass tubes. For voltage-clamp experiments to record miniature inhibitory postsynaptic currents (mIPSCs), patch electrodes were filled with Cs-based internal solution containing (in mM) 110 CsCl, 30 K-gluconate, 1.1 EGTA, 10 HEPES, 0.1 CaCl$_2$, 4 Mg-ATP, 0.3 Na-GTP, and 4 N-(2,6-Dimethylphenylcarbamoylmethyl) triethylammonium bromide (QX-314; Tocris-Cookson, Ellisville, MO); pH = 7.3 (adjusted with CsOH, 280 mOsm). To block glutamatergic EPSCs, 2 µM NBQX (6-cyano-7-nitroquinoxaline-2,3-dione) and 2 µM CPP ((RS)−3-(2-Carboxypiperazin-4-yl)-propyl-1-phosphonic acid),) were added to the bath. Action potential (AP) firing was suppressed by adding 1 µM tetrodotoxin (TTX) to the aCSF. For current-clamp experiments and for recordings of spontaneous postsynaptic excitatory currents (sEPSCs), patch electrodes were filled with K-gluconate-based internal solution containing (in mM) 125 K-gluconate, 20 KCl, 0.2 EGTA, 2 MgCl$_2$, 10 HEPES, and 2 Na$_2$ATP; pH = 7.3 (adjusted with KCl, 280 mOsm). To block GABAergic EPSCs, 25 µM bicuculline methiodide (Tocris-Cookson, Ellisville, MO) was added to the bath. Firing thresholds were estimated from AP phase-plane plots and corrected for a measured liquid junction potential of 7.9 mV. To monitor series resistance on-line and to allow offline estimation of whole-cell membrane resistance and membrane capacitance, a voltage step (10 mV amplitude, 20 msec duration) was delivered at the beginning of each sweep during whole-cell voltage-clamp experiments. Capacitive current transients were analyzed using a simplified two-compartment equivalent circuit model[56]. Mean current transients obtained by averaging ≥ 30 consecutive sweeps were fitted using a double-exponential function $I(t) = A_1 \times e^{-t/\tau_1} + A_2 \times e^{-t/\tau_2} + A_\infty$, where $I(t)$ is the amplitude of the current at time $t$, $A_1$ and $\tau_1$ denote the amplitude and time constant of the fast component of decay, $A_2$ and $\tau_2$ represent the amplitude and time constant of the slower component of decay, and $A_\infty$ is the difference between the holding current and the final steady-state current at the end of the depolarizing pulse. The holding potential for whole-cell voltage-clamp recordings was set to −70 mV. Whole-cell voltage-clamp recordings were included in the analysis if the access resistance was initially ≤ 13 MΩ and did not change by more than 20% during the recording. Recordings with a leak current > 200 pA were rejected. Data were acquired with Patchmaster software (HEKA Elektronik, Germany), low-pass filtered at cut-off frequency of 5 kHz using a Bessel filter and digitized at 20 kHz. All offline analysis was performed with IgorPro (Wavemetrics, USA). mIPSCs were detected using a sliding template-matching algorithm implemented in IgorPro[57].

**Generation and stereotaxic injection of AAV.** To generate shRNA-expressing AAV particles, IgsF9b and control shRNA sequences were first cloned into an AAV-shRNA-GFP vector[58] generously provided by Dr. Ralph DiLeone (Yale University). The following shRNA sequences were used (modified from ref. [14]): IgsF9b, TCATCAAGTTTGGCTACTAT; control (point mutant lacking knock-down activity), TCAT<u>A</u>AGTTC<u>G</u>GCTACTAT. To confirm efficacy of knock-down, the resulting plasmids were co-transfected into HEK cells with a Myc-IgsF9b construct[14] generously provided by Dr. Eunjoon Kim (Korea Advanced Institute of Science and Technology), and IgsF9b levels were quantified using standard immunoblotting procedures[6] (primary antibody: rabbit anti-IgsF9b, Sigma, diluted 1:1000). AAV particles were generated using the pDPrs1/pDPrs2 packaging system (PlasmidFactory, Bielefeld, Germany) and the AAV-shRNA-GFP vectors described above. Plasmids were transfected into HEK cells using calcium phosphate transfection, and virus particles were harvested 72 h later[58,59]. To this end, HEK cells were lysed for 30 min at 37 °C in 20 mM Tris, pH 8.0, containing 150 mM NaCl, 0.5% sodium deoxycholate and benzonase, followed by incubation in 1 M NaCl at 56 °C for 30 min. Lysates were stored at −80 °C overnight, thawed, and purified on a 15%/25%/40%/54% iodixanol gradient by ultracentrifugation (90 min at 370,000× g). The 40% fraction was collected, diluted in PBS containing 1 mM MgCl$_2$ and 2.5 mM KCl, and concentrated using an Amicon 100 K MWCO filter.

Surgical procedures were exactly as described for the injection of Retrobeads, except that 1 µl of virus was injected bilaterally into CeA using a Nanoject II Microinjector (Drummond, Broomall, PA, USA) and a Micro pump controller (WPI). Mice were alternately assigned to receive AAV-control shRNA or AAV-IgsF9b-shRNA injections based on order of birth. The following coordinates relative to Bregma were used: AP (anteroposterior) −0.58, ML (mediolateral) ±2.48, DV (dorsoventral) and −5.4. After surgery, mice were housed in pairs and were allowed to recover for 6 weeks before assessment of behavior in the OF as described above. Mice were sacrificed following OF exposure for verification of the injection site as defined by GFP expression (Supplementary Fig. 5). Only mice in which both bilateral injection sites were correctly positioned in CeM were included in the study. Moreover, mice with any GFP expression in BLA or CeL were excluded, although minor, low-expression leakage into other border areas was tolerated. In total, 11 mice were excluded due to mistargeting (WT + Ctrl shRNA, 4 animals; WT + IgsF9b shRNA, 1 animal; Nlgn2 KO + Ctrl shRNA, 3 animals; and Nlgn2 KO + IgsF9b shRNA, 3 animals).

**Statistical Analysis.** Sample sizes were estimated based on prior experience with the methods used in this study[6,8,60]. All data were analyzed statistically using Prism (GraphPad Software, La Jolla, CA, USA) or Matlab. Outliers were identified using the Grubb's test and were removed prior to statistical analysis. Behavioral scores were subjected to two-way ANOVA with post-hoc Tukey's tests for comparison between groups. Data obtained from histological experiments were analyzed using two-way ANOVA with post-hoc paired, two tailed Student's t-tests for comparison between groups. Data obtained from in vitro electrophysiological experiments were analyzed using two-way ANOVA with post-hoc Tukey's test for comparison between groups. Distributions of mean mIPSC frequencies and amplitudes were analyzed using the Kolmogorov–Smirnov test. Data obtained from in vivo electrophysiological experiments were analyzed using two-way ANOVA with post-hoc Tukey's test for comparisons between groups.

**Code availability.** Custom MATLAB scripts written for the analysis of LFPs are available from the corresponding author upon request.

## Data availability

All data produced from this study are available from the corresponding author upon request.

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

## Acknowledgements

This study was supported by the European Commission (Marie Curie IRG, D.K.-B.; EU-AIMS, N.B.,), the Brain & Behavior Foundation (NARSAD Young Investigator Grant, D.K.-B.), the Deutsche Forschungsgesellschaft (CNMPB and SFB1190/P10, N.B.), the Bundesministerium für Bildung und Forschung (ERA-NET Neuron Synpathy, N.B.) and the Alexander von Humboldt Foundation (Research Fellowship, D.K.-B.). O.B. was a student of the Göttingen Graduate School of Neurosciences and Molecular Biosciences (GGNB) and was funded by a PhD fellowship from the Minerva Foundation. C.P.C. was a student of the Neurasmus Master program and was supported by an Erasmus Mundus scholarship (European Commission). The authors thank Dr. Frédérique Varoqueaux for generating the IgSF9b x Nlgn2 double KO mouse line; Dr. Eunjoon Kim for providing the IgSF9b-myc construct; Dr. Jean-Marc Fritschy for providing the GABA$_A$Rγ2 antibody; Dr. Ralph DiLeone for providing the shRNA plasmid; Dr. Atsushi Miyawaki for providing the pCS-Venus construct; and Dr. Andreas Lüthi for assistance with the establishment of slice recordings in adult mouse amygdala. The authors are grateful to Fritz Benseler, the AGCT Lab, Anja Ronnenberg, the MPIEM animal facility, the Feinmechanik. and the Haustechnik for excellent technical support.

## Author contributions

D.K.-B., O.B., and N.B. conceived the study. D.K.-B. and O.B. designed all experiments (except in vivo electrophysiology experiments, which were designed by H.C.-S. and O.B.). O.B. performed and analyzed most of the behavioral, immunohistochemical, and in vitro electrophysiology experiments and performed stereotaxic surgeries under supervision of D.K.-B. H.C.-S. performed and analyzed in vivo electrophysiology experiments. C.P.C. assisted with stereotaxic surgeries and behavior experiments. M.H. assisted with immunohistochemical experiments and with establishing the in vitro electrophysiology protocol. S.W. assisted with AAV production and with behavioral experiments. H.A. assisted with cFOS quantification. N.K. and L.d.H. assisted with establishing the in vivo electrophysiology protocol. O.M.S. assisted with establishing a protocol for stereotaxic surgery and AAV virus production. Y.Y. provided research reagents. H.E. provided guidance and equipment for behavioral experiments. H.T. assisted with establishing the in vitro electrophysiology experiments and provided software for data analysis. N.B.

provided guidance and support for all experiments. D.K.-B., O.B., and H.C.-S. wrote the paper, and all authors edited and approved the final manuscript.

## Additional information

**Competing interests:** The authors declare no competing interests.

