## [Peer Review File · Nature Communications]

Reviewers' comments:

Reviewer #1 (Remarks to the Author):

What are the major claims of the paper?

Babaev et al. studied the role of IgSF9b, an adhesion protein at inhibitory synapses, associated with affective disorders, in regulating anxiety. They discovered that IgSF9b functionally interacted with Nlgn2 in the amygdala anxiety circuitry. In particular, deleting IgSF9b ameliorate elevated anxiety behavior caused by Nlgn2 deletion. Interestingly, this amelioration is mediated by the circuit level normalization of the over-excitation of the BA-CeM pathway, tested with in vivo electrophysiology, slice physiology and immunohistochemistry. They suggested that the normalization is likely due to enhanced inhibitory synaptic input in CeM, which inhibits the over-excitation of the BA-CeM pathway. More importantly, the behavioral phenotype of Nlgn2 KO can be rescued using the CeM-specific IgSF9b knockdown, arguing that the specific role of IgSF9b in regulating local CeM activity in anxiety. Overall, the MS presents an exciting new line of evidence suggesting that different synapses in the brain circuits are regulated differently based upon their genetic composition which influences the overall circuit function underlying certain behavior. The data were collected and analyzed in a rigorous manner, and the MS is clearly written. This reviewer only has some minor concerns about the data presentation, and discussion.

Are the claims novel?

There are several novel findings in this paper:

1. IgSF9b is a novel schizophrenia-associated gene. Very little is known about its cellular function. This is the first study showing the effect of deleting IgSF9b on animal behavior and how this relates to circuit and cellular alteration.
2. Although the previous literature (only one paper) postulated that IgSF9b acts in a complex with Nlgn2, the current study shows differential functionality of IgSF9b and Nlgn2 in different regions in the amygdala circuitry, i.e. non-cell/synapse autonomous interaction.
3. They discovered that the power of the beta band in CeM during the exploration of the OF center and SAP, was elevated in the Nlgn2 KOs, which was also normalized in the double Kos, correlated with their behavioral rescue.

Will the paper be of interest to others in the field?

This MS will have broad appeal to researchers who are interested in affective disorders, amygdala circuitry and cellular, molecular mechanisms of inhibitory synapse assembly and regulation.

Will the paper influence thinking in the field?

The paper presents evidence of genetic interaction at a specific circuit with behavioral consequences, which provides a new angle how we should think about therapeutic development for complex brain disorders. Furthermore, the idea that inhibitory synapses to inhibitory neurons vs inhibitory synapses to excitatory neurons are regulated differently using different molecular machinery is interesting. The data in this paper provides some evidence suggesting that IgSF9b and Nlgn2 are involved in this differential regulation. It will open up new directions to further address this idea using the tools developed in this paper.

Are the claims convincing? If not, what further evidence is needed?

1. The behavioral rescue is very nicely correlated with the immunohistochemistry data showing the normalization of CeM activity with double KO. It is also correlated with the normalization of the beta band power.
2. Synaptic physiology data suggests that Nlgn2 KO significantly influence BA inhibitory synapses event frequency and quatal size; whereas IgSF9b KO significantly influence CeM inhibitory event frequency. Although the double KO did not show significant increase in mini frequency relative to wt, the accumulative distribution did show significant rightward shift, indicating the IgSF9b-driven effect on mini event frequency. These results support the authors' claim that IgSF9b and Nlgn2 have independent roles in regulating inhibitory synapses.

Improvements on Analysis

Fig. 4b, it is unclear whether the individual data points in the bar graph are numbers of image planes, or number of animal. Animals should be the biological replicates in this analysis.

Fig. 5c&i. there seem to be changes in the variance/distribution with the genetic manipulations on mini frequency, which is somewhat evident in the cumulative prob plot (d&j), and implicative of differential regulation on different neuronal population. It will be useful to present the insets using box plots or violin plots instead of bar graphs to better illustrate the distribution of the data.

Are there other experiments that would strengthen the paper further?

It will be good to know whether shIgSF9b infected CeM neurons show higher frequency of inhibitory synaptic event in wt animals and in Nlgn2 KO animals; and further if this is the case, whether this is cell autonomous. This will enhance our understanding about the IgSF6b deletion based rescue.

Optional

The MS can benefit tremendously if the CeM recordings can be done in a cell-type specific way, or if the different local inhibitory neuron to CeM projection neuron pathways can be targeted to show its effect in terms of rescuing Nlgn2 KO behavioral deficit. However, this may be beyond the scope of the discovery stage of the paper.

Discussion

The discussion is comprehensive, and addresses the main issues of the MS. It would be important to further elaborate on the deficit of beta band oscillation in the Nlgn2 KO, and how it might be involved in affective disorders, and how the IgSF6b KO rescues it via circuit normalization.

Reviewer #2 (Remarks to the Author):

Babaev et al et al., investigated anxiety behavior in several related mutant mouse lines and discovered that genetic deletion of IgSF9b, a key inhibitory synapse adhesion molecule, can reverse prominent anxious phenotypes in neuroligin2 KOs, another inhibitory synapse adhesion molecule. The authors further made effects to work on circuit and synaptic mechanisms that help provide mechanistic insights into the normalization of anxiety-related behaviors in NL2 KO mice by genetic deletion of IgSF9b. It is also important that the authors have identified a critical role of IgSF9b-regulating inhibitory synapses inside CeM in anxiety.

Overall the experiments were well executed, the data are convincing, provide novel insights into the role of IgSF9b-regulating inhibitory synapses in anxiety, and will be of interest to the general neuroscience community, and the statistic analysis is also adequate. Below are specific comments:

1. the title is a bit misleading. There is no evidence supporting a role of inhibitory synapses in amygdala for anxiety-related behaviors related to NL2 in this manuscript. It is a global knockout for NL2. For IgSF9b, the authors have used the local injection of shRNA inside CeM, which supports a role of IgSF9b-related amygdala inhibitory synapses in anxiety.
2. Woo et al., (J of Cell Biology 2013) showed that IgSF9B promotes GABAergic synapse development in hippocampal interneurons, and knockdown of IgSF9B reduces GABAergic transmission in interneurons. Here in the current work, this paper shows that IgSF9B KO increased mIPSCs in CeM neurons. It is important to discuss these different synaptic phenotypes in two different neurons after downregulating IgSF9b expression. What could underlie the different synaptic phenotypes? Similarly, the vGAT staining in Fig6 in IgSF9b KO is also increased, which is

opposite to what were reported in Woo et al. These should be discussed, too.

3. it is commendable that the authors used the local IgSF9b shRNA viral injection in CeM to study the role of IgSF9b in CeM neurons in anxiety (Fig4). However, it is important to examine GABAergic transmission in IgSF9b knockdown neurons in CeM. Are mIPSCs increased (frequency or amplitude) in these knockdown neurons? Similar to KOs? These data should be provided and will strengthen the overall conclusion.

4. for immunohistochemistry in Fig4b, the label indicates the number of puncta/um. Should it be the number of puncta/um² (area)? In Fig6, how do the authors measure the length of the cell body perimeter as there is no reporter (such as GFP) filling up the cell body to delineate the cell body?

5. in Fig6g-h, the VIAAT, but not gephyrin, staining is increased in IgSF9b KO or double KO. Does this suggest that some VIAAT-positive presynaptic terminals in these KOs align with functional postsynapses (as mIPSC frequency is increased) without gephyrin? In addition, can the authors speculate why there are GABAAR subunit-specific effect in different KOs (fig6)?

6. for in vitro electrophysiology (line 986), the aCSF appears to be wrong. It does not contain NaCl and KCl. What is the holding potential when recording mIPSCs?

Reviewer #3 (Remarks to the Author):

This manuscript titled "IgSF9b and Neuroligin2 bidirectionally regulate anxiety-related behaviors 2 through differential effects on amygdala inhibitory synapses" demonstrates a novel role for IgSF9b in regulating anxiety-like behaviors in mice. The authors combine mouse genetics, viral knockdown, behavioral assays, immunohistochemical, and in vivo and in vitro electrophysiological approaches to shed light on how this protein is important to inhibitory synapse function and its relevance to anxiety-like behaviors in the EPM and open field. Overall, the manuscript is interesting and well written, the experiments are performed to a high level of rigor, and the results are appropriately interpreted and discussed within the context of the existing literature. Moreover, the conclusions regarding specific effects of Nlgn2 and IgSF9b on inhibitory transmission in the BA and CeA are well-supported by consistent findings of synaptic puncta labeling and slice electrophysiology. This is a well-composed study. However, there are concerns that should be addressed before the paper will be appropriate for publication.

Major comments:

1. Figure 2b: Please include images that have not been cropped to anatomic landmarks. It is important to show the degree to which Fos labeling differences were specific to the BLA/CeA. It is not typical to show examples of isolated structures as primary data.

2. Figure 2g: the right image in the panel is misleading. Please eliminate the border outlines and show uncropped example images, with insets if necessary. As constructed, the quality of the data are difficult to ascertain and the image does not appear to be scaled appropriately.

3. Figure 4a. Unclear what left-most lane is (I am assuming it is the protein ladder). Please label the size of the ladder fragments shown and where the expected IgSF9b band would be. In the blot, the IgSF9b band is spanning a large range of molecular weights. This raises concern over specificity of antibody and/or virus.

4. Line 240-243 and Figure 5: In each brain region, it would be informative to record both mIPSCs and mEPSCs because both are determinates of excitability and any direct or indirect effects of Nlgn2/IgSF9b on glutamate synapses would be important to consider.

5. It is always difficult to be confident about the specificity of viral targeting to small, deep brain structures, particularly when they are contiguous. It would therefore be valuable to confirm that shRNA injection of a different target (i.e. the CeL or BLA) does not produce the same effect as CeM injection (Fig. 5). Also, can the authors state how many animals were excluded from this experiment due to mistargeting?

Minor comments:

1. Please discuss the type of signaling in which IgSF9b may participate. I.e. what are the downstream consequences of IgSF9b/NIgn2 binding (or lack thereof) that may promote these behavior phenotypes.

2. Figure 5: Did the authors notice any changes in mini kinetics? This could reflect changes in GABA receptor composition.

RESPONSE TO REVIEWERS

IgSF9b regulates anxiety-related behaviors through effects on centromedial amygdala inhibitory synapses

Olga Babaev, Hugo Cruces-Solis, Carolina Piletti Chatain, Matthieu Hammer, Sally Wenger, Heba Ali, Nikolaos Karalis, Livia de Hoz, Oliver M. Schlüter, Yuchio Yanagawa, Hannelore Ehrenreich, Holger Taschenberger, Nils Brose and Dilja Krueger-Burg

Dear Dr. Carr,

Encouraged by the overall positive evaluations of the three reviewers, we would like to submit the attached revised manuscript for your consideration. We have addressed each of the points made by the reviewers as detailed below, and have added several experiments and clarifications following their suggestions. All of the changes in the manuscript and supplement have been highlighted in yellow.

We thank the reviewers for their insightful comments and helpful suggestions, which have substantially improved the manuscript, and we hope that the revised manuscript will now be acceptable for publication in Nature Communications.

Sincerely,
Dilja Krueger-Burg

Reviewer #1

What are the major claims of the paper?

Babaev et al. studied the role of IgSF9b, an adhesion protein at inhibitory synapses, associated with affective disorders, in regulating anxiety. They discovered that IgSF9b functionally interacted with Nlgn2 in the amygdala anxiety circuitry. In particular, deleting IgSF9b ameliorate elevated anxiety behavior caused by Nlgn2 deletion. Interestingly, this amelioration is mediated by the circuit level normalization of the over-excitation of the BA-CeM pathway, tested with in vivo electrophysiology, slice physiology and immunohistochemistry. They suggested that the normalization is likely due to enhanced inhibitory synaptic input in CeM, which inhibits the over-excitation of the BA-CeM pathway. More importantly, the behavioral phenotype of Nlgn2 KO can be rescued using the CeM-specific IgSF9b knockdown, arguing that the specific role of IgSF9b in regulating local CeM activity in anxiety. Overall, the MS presents an exciting new line of evidence suggesting that different synapses in the brain circuits are regulated differently based upon their genetic composition which influences the overall circuit function underlying certain behavior. The data were collected and analyzed in a rigorous manner, and the MS is clearly written. This reviewer only has some minor concerns about the data presentation, and discussion.

Are the claims novel?

There are several novel findings in this paper:

1. *IgSF9b is a novel schizophrenia-associated gene. Very little is known about its cellular function. This is the first study showing the effect of deleting IgSF9b on animal behavior and how this relates to circuit and cellular alteration.*

2. *Although the previous literature (only one paper) postulated that IgSF9b acts in a complex with Nlgn2, the current study shows differential functionality of IgSF9b and Nlgn2 in different regions in the amygdala circuitry, i.e. non-cell/synapse autonomous interaction.*

3. *They discovered that the power of the beta band in CeM during the exploration of the OF center and SAP, was elevated in the Nlgn2 KOs, which was also normalized in the double KOs, correlated with their behavioral rescue.*

Will the paper be of interest to others in the field?

This MS will have broad appeal to researchers who are interested in affective disorders, amygdala circuitry and cellular, molecular mechanisms of inhibitory synapse assembly and regulation.

Will the paper influence thinking in the field?

The paper presents evidence of genetic interaction at a specific circuit with behavioral consequences, which provides a new angle how we should think about therapeutic development for complex brain disorders. Furthermore, the idea that inhibitory synapses to inhibitory neurons vs inhibitory synapses to excitatory neurons are regulated differently using different molecular machinery is interesting. The data in this paper provides some evidence suggesting that IgSF9b and Nlgn2 are involved in this differential regulation. It will open up new directions to further address this idea using the tools developed in this paper.

Are the claims convincing? If not, what further evidence is needed?

1. *The behavioral rescue is very nicely correlated with the immunohistochemistry data showing the normalization of CeM activity with double KO. It is also correlated with the normalization of the beta band power.*

2. *Synaptic physiology data suggests that Nlgn2 KO significantly influence BA inhibitory synapses event frequency and quatal size; whereas IgSF9b KO significantly influence CeM inhibitory event frequency. Although the double KO did not show significant increase in mini frequency relative to wt, the accumulative distribution did show significant rightward shift, indicating the IgSF9b-driven effect on mini event frequency. These results support the authors' claim that IgSF9b and Nlgn2 have independent roles in regulating inhibitory synapses.*

Improvements on Analysis:

Fig. 4b, it is unclear whether the individual data points in the bar graph are numbers of image planes, or number of animal. Animals should be the biological replicates in this analysis.

This graph previously indeed was based on the number of image planes from 2 animals per group. We have now increased the n to 3 and have used these three animals as the biological replicates as suggested (Figure 4b).

Fig. 5c&i. there seem to be changes in the variance/distribution with the genetic manipulations on mini frequency, which is somewhat evident in the cumulative prob plot (d&j), and implicative of differential regulation on different neuronal population. It will be useful to present the insets using box plots or violin plots instead of bar graphs to better illustrate the distribution of the data.

We agree with the reviewer that the distribution of the average mIPSC frequency seems different across genotypes, and to illustrate this we plotted the cumulative distributions of the average frequency and amplitude of each cell recorded for each genotype (Fig. 5d and j in the original manuscript as mentioned; Fig. 5i and o in the current version). To further illustrate this variance we have now replaced the bar graphs in the insets with box plots as requested (Fig. 5i, k, o, q; Fig 7d, f).

Are there other experiments that would strengthen the paper further?

It will be good to know whether shIgSF9b infected CeM neurons show higher frequency of inhibitory synaptic event in wt animals and in Nlgn2 KO animals; and further if this is the case, whether this is cell autonomous. This will enhance our understanding about the IgSF6b deletion based rescue.

We thank the reviewer for this important suggestion. For technical reasons resulting from the advanced age of the animals in this experiment, we were unfortunately unable to perform electrophysiology in AAV-shRNA-injection animals. Patch clamp recordings from adult animals in the amygdala are challenging even in the 8-10 week old mice used in our other experiments, and at 14-15 weeks, the age that inevitably results from the 6-week delay following AAV injection, the cells were no longer viable long enough for reliable recordings. However, since we agree with the reviewer that this is an important point which will substantially strengthen our model, we have instead performed the suggested experiment looking at staining for the vesicular inhibitory amino acid transporter VIAAT, which is upregulated in the constitutive IgSF9b KO and double KO (Fig. 6g of the original manuscript, now in Fig. 6c), and which we believe to represent the molecular basis for the increase in mIPSC frequency. In this new experiment, we have now shown that IgSF9b shRNA-infected CeM neurons show the same upregulation of perisomatic VIAAT puncta compared to control shRNA-infected neurons, with an increase in the size or number of VIAAT-positive puncta observed in WT or Nlgn2 KO neurons infected with IgSF9b shRNA, respectively. These data, which confirm that shRNA-based deletion of IgSF9b indeed mimics the mechanistic basis of the constitutive KO, have been added to Figure 6 (Fig. 6h-j). To accommodate these and other additional data, and to slightly shift the focus of the manuscript as described below, we have split the original Figures 5 and 6 into three figures, now Figures 5-7.

Since many of the inhibitory VIAAT-positive presynaptic terminals in the CeM originate from cell bodies outside of the CeM, which would not have been affected by the shRNA, we conclude that our effect is likely a cell-autonomous postsynaptic effect of IgSF9b knock-down in the targeted neurons. However, we cannot exclude that part of the effect might originate from the presynaptic terminals of local inhibitory interneurons. Since IgSF9b is likely to be a homophilic adhesion molecule, which is required on both the pre- and the postsynaptic terminal, it could be expected that deletion of IgSF9b in the pre- or the postsynaptic neuron would have the same consequences. This is an interesting question that we have now addressed in our discussion (p12, line 358-363) and are planning to investigate experimentally in future studies: “Together with the observation that IgSF9b, like its mammalian homolog IgSF9 and its *Drosophila* homolog Turtle, forms primarily or exclusively homophilic interactions^{14, 16, 42}, our data indicate that IgSF9b provides a transsynaptic signaling complex that regulates the organization of the inhibitory presynaptic terminal. By this logic, deletion of IgSF9b from either the pre- or the postsynaptic neuron may result in rearrangements of the presynaptic terminal”

Optional

The MS can benefit tremendously if the CeM recordings can be done in a cell-type specific way, or if the different local inhibitory neuron to CeM projection neuron pathways can be targeted to show its effect in terms of rescuing Nlgn2 KO behavioral deficit. However, this may be beyond the scope of the discovery stage of the paper.

We agree with the reviewer that experiments to look at cell type specificity are extremely interesting, and indeed these experiments are underway. However, this constitutes an entirely new study and, as the reviewer has noted, is unfortunately beyond the scope of the present manuscript. To address this important issue, we have now added a comment to this effect in the discussion (p13, lines 395-396): “Cell type- and circuit-specific analysis of IgSF9b function will be essential for fully understanding its role in the brain”.

Discussion

The discussion is comprehensive, and addresses the main issues of the MS. It would be important to further elaborate on the deficit of beta band oscillation in the Nlgn2 KO, and how it might be involved in affective disorders, and how the IgSF6b KO rescues it via circuit normalization.

This is indeed a very interesting question, albeit one that we cannot answer definitively at this point. Surprisingly little is known about either the function of beta oscillations in network information processing or the cellular and synaptic mechanisms that underlie the generation of this oscillatory activity, unlike e.g. gamma oscillations which are very well characterized both functionally and mechanistically. We have therefore now added the following statement to the discussion to clarify this issue (p14, lines 439-452):

“At present both the function of these beta oscillations in anxiety processing and the mechanisms that underlie their modulation by Nlgn2 and IgSF9b remain unknown. Beta oscillations have previously been observed in the basal ganglia and somatosensory cortex during decision-making tasks^{47, 48}, in the hippocampus upon exposure to novelty, and in the mediodorsal thalamus during working memory-related tasks^{49, 50}, highlighting their multifaceted role in information processing. Not surprisingly, therefore, alterations in beta oscillatory power have been linked to multiple disease states, most notably Parkinson’s disease⁵¹, but also anxiety disorders and others^{52, 53}. However, their precise function in mediating these behavioral processes remains controversial and an area of active investigation. Similarly, the cellular and synaptic mechanisms leading to the generation of beta rhythms are largely unknown, although changes in inhibitory synaptic activity are known to affect local field potentials in general⁵⁴ and beta activity in specific. Further experiments will be essential to fully understand how Nlgn2 and IgSF9b regulate beta oscillations in the CeM and how this contributes to the generation and normalization of pathological anxiety processing.”

Reviewer #2

Babaev et al et al., investigated anxiety behavior in several related mutant mouse lines and discovered that genetic deletion of IgSF9b, a key inhibitory synapse adhesion molecule, can reverse prominent anxious phenotypes in neuroligin2 KOs, another inhibitory synapse adhesion molecule. The authors further made efforts to work on circuit and synaptic mechanisms that help provide mechanistic insights into the normalization of anxiety-related behaviors in NL2 KO mice by genetic deletion of IgSF9b. It is also important that the authors

have identified a critical role of IgSF9b-regulating inhibitory synapses inside CeM in anxiety.

Overall the experiments were well executed, the data are convincing, provide novel insights into the role of IgSF9b-regulating inhibitory synapses in anxiety, and will be of interest to the general neuroscience community, and the statistic analysis is also adequate. Below are specific comments:

(1) The title is a bit misleading. There is no evidence supporting a role of inhibitory synapses in amygdala for anxiety-related behaviors related to NL2 in this manuscript. It is a global knockout for NL2. For IgSF9b, the authors have used the local injection of shRNA inside CeM, which supports a role of IgSF9b-related amygdala inhibitory synapses in anxiety.

We agree with this assessment, particularly in the revised manuscript, which focuses even more on the role of IgSF9b in the CeM. We have therefore changed the title to read “IgSF9b regulates anxiety-related behaviors through effects on centromedial amygdala inhibitory synapses”.

(2) Woo et al., (J of Cell Biology 2013) showed that IgSF9B promotes GABAergic synapse development in hippocampal interneurons, and knockdown of IgSF9B reduces GABAergic transmission in interneurons. Here in the current work, this paper shows that IgSF9B KO increased mIPSCs in CeM neurons. It is important to discuss these different synaptic phenotypes in two different neurons after downregulating IgSF9b expression. What could underlie the different synaptic phenotypes? Similarly, the vGAT staining in Fig6 in IgSF9b KO is also increased, which is opposite to what were reported in Woo et al. These should be discussed, too.

This is a very interesting issue, and one that likely reflects the different roles of IgSF9b in various neuronal populations. While we cannot definitively explain the differences between the Woo et al. study (ref. 14 in our revised manuscript) and our findings, we have now included the following paragraph in our discussion (p13, lines 386-396):

“Moreover, the function of IgSF9b at inhibitory synapses may depend on the identity of the pre- and/or postsynaptic inhibitory neurons. In hippocampal cultures, shRNA-mediated deletion of IgSF9b resulted in a decrease in mIPSC frequency and a reduction in VIAAT-positive gephyrin clusters, in contrast to the increase in mIPSC frequency and VIAAT puncta observed here in the CeM (Fig. 5-6). Since the inhibitory neuron subtypes found in the CeM are entirely distinct from those observed in the hippocampus, it is conceivable that these different subtypes may differentially employ IgSF9b at inhibitory postsynapses. Alternatively, IgSF9b may play distinct roles at different stages of synaptic development (i.e. at developing synapses in neuronal cultures vs. mature synapses in adult mice) or in dissociated cultures vs. intact neuronal networks. Cell type- and circuit-specific analysis of IgSF9b function will be essential for fully understanding its role in the brain”.

(3) It is commendable that the authors used the local IgSF9b shRNA viral injection in CeM to study the role of IgSF9b in CeM neurons in anxiety (Fig4). However, it is important to examine GABAergic transmission in IgSF9b knockdown neurons in CeM. Are mIPSCs increased (frequency or amplitude) in these knockdown neurons? Similar to KOs? These data should be provided and will strengthen the overall conclusion.

We thank the reviewer for this important point, which was also raised by reviewer 1. As described in detail above (reviewer 1, comment 3), we were unable to record reliable mIPSCs

in the 14-15 week old mice that were the inevitable consequence of the 6-week delay following AAV injection. However, we added data to show that IgSF9b shRNA injection indeed leads to an increase in VIAAT puncta, which is also observed in constitutive IgSF9b deletion and is likely the molecular basis of the increase in mIPSC frequency. These data have been added to Figure 6 e-g and strengthen the mechanistic basis of our model.

(4) For immunohistochemistry in Fig4b, the label indicates the number of puncta/um. Should it be the number of puncta/um² (area)? In Fig6, how do the authors measure the length of the cell body perimeter as there is no reporter (such as GFP) filling up the cell body to delineate the cell body?

The quantification of IgSF9b levels in Figure 4b was conducted on perisomatic IgSF9b puncta, to be consistent with the remaining immunohistochemical analyses in the study. The label ‘number of puncta/μm’ is therefore correct, since it refers to the number of puncta per length of the cell perimeter. The cell body perimeter for all immunohistochemistry experiments was defined as follows, consistent with our previous studies (Hammer et al. 2015, Babaev et al. 2016): “To quantify perisomatic synapses, the perisomatic area was identified by manually tracing the perimeter of the cell body (defined as a circular area largely void of immunofluorescence) as previously described. The perimeter was then expanded by 1.4 μm or 2 μm in each direction for quantification of postsynaptic puncta or presynaptic puncta, respectively.” This statement has been added to the manuscript on p23, lines 727-731.

(5) In Fig6g-h, the VIAAT, but not gephyrin, staining is increased in IgSF9b KO or double KO. Does this suggest that some VIAAT-positive presynaptic terminals in these KOs align with functional postsynapses (as mIPSC frequency is increased) without gephyrin? In addition, can the authors speculate why there are GABAAR subunit-specific effect in different KOs (fig6)?

We thank the reviewer for these very interesting questions. Regarding the increase in VIAAT staining without an increase in gephyrin staining, we interpret this as an increase in the number of VIAAT-positive vesicles per terminal, rather than an increase in the number of inhibitory presynaptic terminals. Interestingly, a similar effect was recently observed in IgSF21 KO mice (Tanabe et al. 2017, Nat. Commun.), although here there was a decrease in mIPSC frequency, VIAAT staining and synaptic vesicles. We have now modified the discussion and added the following sentence to address this question (p12, line 355-358): “The increase in mIPSC frequency and VIAAT staining without a concomitant increase in gephyrin indicates that IgSF9b deletion may result in an increase in the number of VIAAT-positive vesicles per synaptic terminal, analogous to effects recently observed in IgSF21 KO mice (albeit with opposite polarity)”.

The issue of GABA_AR subunit-specific effects of IgSF9b deletion is more complex to address and importantly, these effects do not appear to contribute substantially to synaptic transmission in the CeM (as evidenced by the lack of effects of mIPSC amplitude). We have now added a sentence to this effect to the discussion (p12, line 363-367): “Moreover, the differential effects of IgSF9b deletion of GABA_AR α1 and γ2 subunits in the CeM indicate that IgSF9b may also regulate the subunit composition of GABA_ARs at inhibitory postsynapses, although the mechanism or functional significance of this effect remains unknown.”

(6) For in vitro electrophysiology (line 986), the aCSF appears to be wrong. It does not contain NaCl and KCl. What is the holding potential when recording mIPSCs?

We apologize for the confusing description of our aCSF. The basic composition of the aCSF for all experiments is listed in the first paragraph of the section ‘7.1. Preparation of slices’, and it contains the following components: aCSF, 124 mM NaCl, 2.7 mM KCl, 26 mM NaHCO₃ and 1.25 mM NaH₂PO₄ (p31 line 973-4 of the original manuscript, now p23 lines 741-742). For the recordings, we also added CaCl₂, MgCl₂ and other components to the above aCSF, and we originally described this as ‘aCSF containing CaCl₂, MgCl₂ etc’ (p31 line 986 in the original manuscript). However, we agree that this is misleading, and we have now changed this to ‘aCSF with additional CaCl₂, MgCl₂ etc.’ in the current version of the manuscript (p24, line 754). We thank the reviewer for pointing this out.

The holding potential for whole-cell patch clamp recording was set to -70 mV. We have now added this information to p24, line 785.

Reviewer #3

This manuscript titled “IgSF9b and Neuroligin2 bidirectionally regulate anxiety-related behaviors 2 through differential effects on amygdala inhibitory synapses” demonstrates a novel role for IgSF9b in regulating anxiety-like behaviors in mice. The authors combine mouse genetics, viral knockdown, behavioral assays, immunohistochemical, and in vivo and in vitro electrophysiological approaches to shed light on how this protein is important to inhibitory synapse function and its relevance to anxiety-like behaviors in the EPM and open field. Overall, the manuscript is interesting and well written, the experiments are performed to a high level of rigor, and the results are appropriately interpreted and discussed within the context of the existing literature. Moreover, the conclusions regarding specific effects of Nlgn2 and IgSF9b on inhibitory transmission in the BA and CeA are well-supported by consistent findings of synaptic puncta labeling and slice electrophysiology. This is a well-composed study. However, there are concerns that should be addressed before the paper will be appropriate for publication.

Major comments:

(1) Figure 2b: Please include images that have not been cropped to anatomic landmarks. It is important to show the degree to which Fos labeling differences were specific to the BLA/CeA. It is not typical to show examples of isolated structures as primary data.

The images in Fig. 2 were originally cropped in the interest of space, and specificity of the changes in BA and CeM was demonstrated by the fact that no changes were observed in LA and CeL in the same sections (Fig 2c, e). However, we agree with the reviewer that access to the primary data is important, and we have therefore now substantially rearranged Fig. 2 to include the uncropped images as requested. In addition, we have added data to Fig. 2b to show that cFos is not altered basally in any of the genotypes (p5, lines 132-134): “The observed increase in cFos was induced by exposure to the OF, as basal cFos in mice taken directly from their home cages did not differ among genotypes”. Together with the lack of changes in the LA and the CeL, these data confirm that the observed changes in the BA and the CeM are specifically induced by anxiogenic conditions in these regions. We cannot rule out that additional changes occur in other brain regions that were not included in our analysis,

but based on our AAV-shRNA data the changes in the CeM are the most relevant for the anxiolytic effect of IgSF9b deletion.

(2) Figure 2g: the right image in the panel is misleading. Please eliminate the border outlines and show uncropped example images, with insets if necessary. As constructed, the quality of the data are difficult to ascertain and the image does not appear to be scaled appropriately.

We have now included uncropped versions of the example images as requested, and we have included scale bars to indicate the scale of the images. We hope this will help to confirm the quality of the underlying data.

(3) Figure 4a. Unclear what left-most lane is (I am assuming it is the protein ladder). Please label the size of the ladder fragments shown and where the expected IgSF9b band would be. In the blot, the IgSF9b band is spanning a large range of molecular weights. This raises concern over specificity of antibody and/or virus.

The left-most lane in Fig. 4a is indeed the protein ladder – we apologize for the oversight in labeling this lane. We have now added labels to indicate the size of the fragments.

With respect to the multiple bands observed in the immunoblot: IgSF9b is subjected to several posttranslational modifications, including glycosylation and sialylation (Woo et al. 2013, JCB). These modifications influence protein migration on SDS-PAGE gels and cause multiple bands that correspond to differentially modified forms. We have now added data to show that all of these bands are absent in the IgSF9b KO mouse (Supplementary Fig. 6a), confirming the specificity of the antibody for immunoblotting. Moreover, immunohistochemistry using this antibody reveals punctate staining in the WT but not the IgSF9b KO mouse (Supplementary Fig. 6a-b in the original manuscript, now Supplementary Fig. 6b), confirming the specificity of the antibody for immunohistochemistry.

(4) Line 240-243 and Figure 5: In each brain region, it would be informative to record both mIPSCs and mEPSCs because both are determinates of excitability and any direct or indirect effects of Nlgn2/IgSF9b on glutamate synapses would be important to consider.

We thank the reviewer for pointing out this issue, and we agree that it is important to consider any influences on glutamatergic synapses or excitability. We have therefore now added substantial new data to investigate sEPSCs and excitability in Nlgn2 KOs, IgSF9b KOs and double KOs in the CeM (Fig 5c-e, Fig. 5i-q, and Table 1). None of the investigated genotypes affected excitatory synaptic transmission or basal excitability of CeM neurons, indicating that the effects we observe are specific for inhibitory synapses.

Given that the primary focus of our model lies on the role of IgSF9b in the CeM, and in view of the fact that the double KO breeding scheme provides a limited number of animals per genotype, we did not record sEPSCs in the BA. We have now rearranged the manuscript to better reflect the focus on the CeM.

(5) It is always difficult to be confident about the specificity of viral targeting to small, deep brain structures, particularly when they are contiguous. It would therefore be valuable to confirm that shRNA injection of a different target (i.e. the CeL or BLA) does not produce the same effect as CeM injection (Fig. 5). Also, can the authors state how many animals were excluded from this experiment due to mistargeting?

We agree that specifically targeting small structures such as the CeM is challenging, which is why we conducted a thorough histological examination of GFP expression in every animal included in the analysis. Only animals in which injections were bilaterally targeted to the CeM were included, and animals showing any GFP expression in the CeL or the BLA were excluded (WT + Ctrl shRNA, 4 animals; WT + IgSF9b shRNA, 1 animal; Nlgn2 KO + Ctrl shRNA, 3 animals; Nlgn2 KO + IgSF9b shRNA, 3 animals). We have now included this information in our manuscript on p26, lines 823-825. Due to the highly involved nature of this experiment, repeating the experiment in a separate brain region was unfortunately beyond the scope of the current manuscript. Moreover, since IgSF9b is likely a homophilic synaptic adhesion molecule, deleting it from presynaptic inhibitory inputs such those coming from the CeL may have similar mechanistic consequences in the CeM as a local postsynaptic deletion in the CeM directly, and this might therefore not be an adequate control. Indeed, this question of synapse and circuit specificity is an important one, which we are actively pursuing, but which, as reviewer 1 pointed out, is beyond the scope of the current manuscript.

Minor comments:

(1) Please discuss the type of signaling in which IgSF9b may participate. I.e. what are the downstream consequences of IgSF9b/Nlgn2 binding (or lack thereof) that may promote these behavior phenotypes.

Given that IgSF9b deletion results in an increase in mIPSC frequency and presynaptic VIAAT staining but not postsynaptic gephyrin staining, we hypothesize that IgSF9b signals transsynaptically to increase the number of VIAAT-positive synaptic vesicles in the presynaptic terminal, thus strengthening inhibitory synaptic transmission in the CeM. We have now extended our discussion of this mechanism on p11/12, lines 385-398 of the manuscript: “The increases in mIPSC frequency and VIAAT staining without a concomitant increase in gephyrin indicate that IgSF9b deletion may result in an increase in the number of VIAAT-positive vesicles per synaptic terminal, analogous to effects recently observed in IgSF21 KO mice (albeit with opposite polarity)”. As described in our summary figure (Fig. 8), we postulate that this increased inhibition mediates the behavioral normalization of the anxiety phenotype. “Specifically, our data support a model in which reduced inhibition in BA of Nlgn2 KO mice results in overactivation of BA → CeM projection neurons under anxiogenic conditions, which is counteracted in the CeM by the increased inhibition resulting from additional deletion of IgSF9b” (p11, lines 341-344). The precise signaling pathway linking IgSF9b to synaptic vesicle accumulation remains to be determined, as virtually nothing is currently known about IgSF9b signaling pathways, and this will be an interesting avenue to pursue for future studies.

(2) Figure 5: Did the authors notice any changes in mini kinetics? This could reflect changes in GABA receptor composition.

This is an interesting question. We did not see any significant changes in rise time or decay time of mIPSCs or sEPSCs in any of the genotypes investigated. We have now added this information to the manuscript (Table 1).

REVIEWERS' COMMENTS:

Reviewer #1 (Remarks to the Author):

This reviewer is satisfied with the author's response.

Reviewer #2 (Remarks to the Author):

The authors have addressed my concerns and I have no more questions.

Reviewer #3 (Remarks to the Author):

I am satisfied with the response by the authors, and the manuscript is substantially improved.

RESPONSE TO REVIEWERS

IgSF9b regulates anxiety behaviors through effects on centromedial amygdala inhibitory synapses

Olga Babaev, Hugo Cruces-Solis, Carolina Piletti Chatain, Matthieu Hammer, Sally Wenger, Heba Ali, Nikolaos Karalis, Livia de Hoz, Oliver M. Schlüter, Yuchio Yanagawa, Hannelore Ehrenreich, Holger Taschenberger, Nils Brose and Dilja Krueger-Burg

We thank the reviewers for re-reviewing our manuscript and for their comments as listed below. We are very happy to hear that all three reviewers are satisfied with our revisions and have no further concerns.

Reviewer #1 (Remarks to the Author):

This reviewer is satisfied with the author's response.

Reviewer #2 (Remarks to the Author):

The authors have addressed my concerns and I have no more questions.

Reviewer #3 (Remarks to the Author):

I am satisfied with the response by the authors, and the manuscript is substantially improved.